# Ferromagnetism on an atom-thick & extended 2D metal-organic coordination network

Jorge Lobo-Checa [1,2] ✉, Leyre Hernández-López[1,2], Mikhail M. Otrokov [3,4,5,13] ✉, Ignacio Piquero-Zulaica [6], Adriana E. Candia [1,7,8], Pierluigi Gargiani [9], David Serrate [1,2], Fernando Delgado [10], Manuel Valvidares [9], Jorge Cerdá[11], Andrés Arnau[3,4,12] ✉ & Fernando Bartolomé [1,2] ✉

Ferromagnetism is the collective alignment of atomic spins that retain a net magnetic moment below the Curie temperature, even in the absence of external magnetic fields. Reducing this fundamental property into strictly two-dimensions was proposed in metal-organic coordination networks, but thus far has eluded experimental realization. In this work, we demonstrate that extended, cooperative ferromagnetism is feasible in an atomically thin two-dimensional metal-organic coordination network, despite only $\approx 5\%$ of the monolayer being composed of Fe atoms. The resulting ferromagnetic state exhibits an out-of-plane easy-axis square-like hysteresis loop with large coercive fields over 2 Tesla, significant magnetic anisotropy, and persists up to $T_C \approx 35$ K. These properties are driven by exchange interactions mainly mediated by the molecular linkers. Our findings resolve a two decade search for ferromagnetism in two-dimensional metal-organic coordination networks.

The formation of extended two-dimensional (2D) magnetic order has long been an active quest in condensed matter physics. In two dimensions, the Mermin-Wagner theorem precludes the formation of isotropic ferromagnetic order when mediated by short-range exchange interactions at finite temperatures[1]. This initially limited the investigation and exemplary cases to bulk materials featuring dominant in-plane interactions[2,3] and ultrathin inorganic layers supported on metallic surfaces[4–7]. It is only very recently that the first examples of pure (i.e. substrate decoupled), extended 2D-ferromagnetism (FM) were obtained by exfoliation of van der Waals crystals[8,9]. In these cases the Mermin-Wagner limitations were surmounted by the presence of significant magnetic anisotropy. This development immediately sparked widespread attention due to the many envisioned fundamental implications and extensive practical applications[10–12]. However, integrating layered van der Waals materials into devices has

[1]Instituto de Nanociencia y Materiales de Aragón (INMA), CSIC-Universidad de Zaragoza, 50009 Zaragoza, Spain. [2]Departamento de Física de la Materia Condensada, Universidad de Zaragoza, E-50009 Zaragoza, Spain. [3]Centro de Física de Materiales CSIC/UPV-EHU-Materials Physics Center, Manuel Lardizabal 5, E-20018 San Sebastián, Spain. [4]Donostia International Physics Center, Paseo Manuel de Lardizabal 4, E-20018 San Sebastian, Spain. [5]IKERBASQUE, Basque Foundation for Science, E-48011 Bilbao, Spain. [6]Physics Department E20, Technical University of Munich, 85748 Garching, Germany. [7]Instituto de Desarrollo Tecnológico para la Industria Química (INTEC-UNL-CONICET), 3000 Santa Fe, Argentina. [8]Instituto de Física del Litoral, Universidad Nacional del Litoral (IFIS-UNL-CONICET), 3000 Santa Fe, Argentina. [9]ALBA Synchrotron Light Source, E-08290 Cerdanyola del Vallès, Spain. [10]Instituto de Estudios Avanzados IUDEA, Departamento de Física, Universidad de La Laguna, C/Astrofísico Francisco Sánchez, s/n, 38203 La Laguna, Spain. [11]Instituto de Ciencia de Materiales de Madrid, CSIC, Cantoblanco 28049 Madrid, Spain. [12]Departamento de Polímeros y Materiales Avanzados: Física, Química y Tecnología, Facultad de Química UPV/EHU, 20080 Donostia-San Sebastián, Spain. [13]Present address: Instituto de Nanociencia y Materiales de Aragón (INMA), CSIC-Universidad de Zaragoza, Zaragoza 50009, Spain. ✉e-mail: jorge.lobo@csic.es; mikhail.otrokov@csic.es; andres.arnau@ehu.eus; fernando.bartolome@csic.es

turned out to be extremely challenging as the lateral size and exact thickness of these layers is difficult to control[12].

Earlier candidates to exhibit 2D-FM at the single-layer limit were metal-organic coordination networks (MOCNs) grown on metallic supports[13,14]. 2D-MOCNs a-priori contain all the essential ingredients to display 2D-FM: Selectable metallic centers providing non-zero atomic spins and incomplete quenching of the magnetic orbital moment (given their reduced point symmetry and chemical coordination)[15,16], periodic spacing of these magnetic moments over the (non-magnetic) surfaces[17], tunable lateral separation between adatoms by the synthetically variable organic linkers[18], reduced electronic overlap of these metal centers with the substrate after 2D-MOCN formation[19], and technically simple fabrication as they follow self-assembly protocols close to room temperature[20,21]. Despite these features, 2D-MOCNs at the single-layer limit have historically failed to explicitly exhibit 2D ferromagnetic remanence[13,14,17,22–27]. Many previous studies of 2D-MOCNs exhibited noticeable magnetic anisotropies, but at the single-layer limit neither spontaneous magnetization nor remanence were ever observed[13,14,17,22–27]. Moreover, the coupling among these metal centers is generally interpreted in terms of superexchange mechanisms through the organic ligands and less so by surface electrons[28].

In this work, we study the magnetism of a single, atomically thin 2D-MOCN consisting of Fe atom centers and 9,10-dicyanoanthracene (DCA) molecular linkers forming a mixed honeycomb kagome lattice on Au(111). We take advantage of the monodomain and extended character of this network in a multitechnique approach. Particularly, we combine scanning tunneling microscopy and spectroscopy (STM/STS), low energy electron diffraction (LEED), X-Ray absorption spectroscopy (XAS), X-Ray magnetic circular dichroism (XMCD), and X-ray photoemission spectroscopy (XPS) techniques. We present clear evidence of long-range ferromagnetic order in a 2D-MOCN with a Curie

temperature ($T_C$) of ≈ 35 K. The ferromagnetic state displays very strong out-of-plane (OOP) magnetic anisotropy and has a square hysteresis loop with a coercive field of ≈ 2.1 T. The XMCD orbital sum rule yields a maximal unquenched OOP orbital magnetic moment for Fe centers of $\langle m_z^l \rangle \approx 2\,\mu_B$. The magnetization as a function of temperature through the FM phase transition falls close to the honeycomb 2D Ising model involving strong uniaxial anisotropic magnetic centers. These results strongly differ from those obtained for the inorganic system formed in absence of DCA molecules, which results in an array of Fe clusters on the Au(111) surface. We make use of first-principles density functional theory calculations to explain the observation of FM at finite temperature in this 2D-MOCN system, which exhibits a large single ion anisotropy at the Fe atoms and a significant exchange interaction across the molecular linkers with a limited contribution through the underlying substrate.

## Results and discussion
### Spatial and electronic structure of the 2D-MOCN
The self-assembled Fe-DCA lattice is formed by sequential deposition of DCA molecules and Fe atoms on Au(111). Prior to the metal evaporation, the molecules form compact islands (see Supplementary Fig. S1) that evolve into the open network under a stoichiometric Fe:DCA relation of 2:3 (see Supplementary Fig. S2 for the effects of deviation from this proportion). The network is perfected after a mild annealing at 373 K for 10 min, generating large and monodomain network islands, similar to Cu-DCA/Cu(111)[29]. Figure 1a shows a typical overview of this 2D-MOCN, where the Fe centers form a honeycomb array and the DCA linkers a Kagome sublattice (cf. Fig. 1b). This network does not destroy the herringbone reconstruction of the underlying Au(111), suggesting a weak surface-MOCN interaction[17]. The analysis of the LEED patterns (see Fig. 1c and Supplementary Fig. S3e, f) show a hexagonal network with $(4\sqrt{3} \times 4\sqrt{3})R30°$ structure with

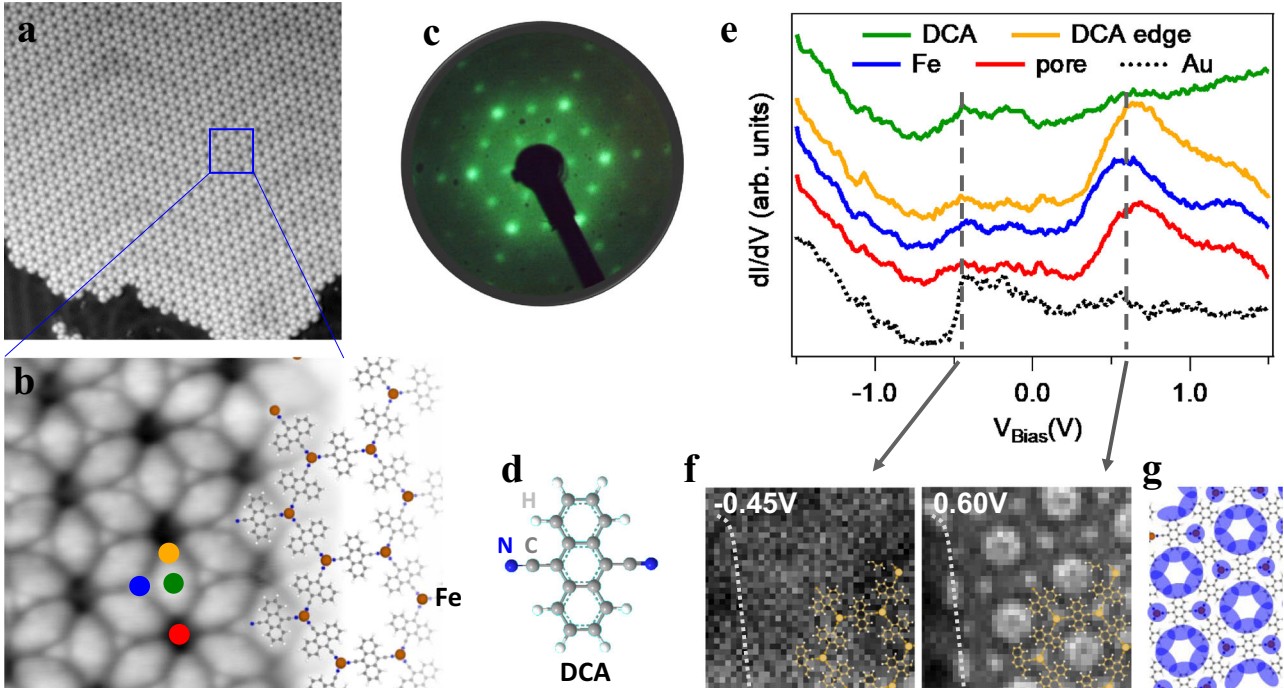

**Fig. 1 | Atomic and electronic structure of the Fe-DCA/Au(111) network.** Overview (**a**) and close-up (**b**) images of the 2D-metal-organic lattice. A model of the MOCN structure is overlaid in **b** with the molecules forming a kagome substructure and the Fe centers (orange-brown spheres) a honeycomb sublattice. **c** LEED pattern of this network (at 19 eV and room temperature) displaying a $(4\sqrt{3} \times 4\sqrt{3})R30°$ structure with respect to the underlying substrate. **d** Stick-ball model of 9,10-dicyanoanthracene (DCA) molecule (C in gray, N in blue and H in white). **e** STS averaged spectra extracted from a dI/dV grid at the positions marked as circles of the same color in **b**. A network (collective) electronic state is located around 0.60 eV. **f** dI/dV isoenergetic maps at the energies indicated by the vertical discontinuous lines in **e**. The gray dotted line on the left of the maps marks the edge of an island. **g** Cartoon of the dominant spatial distribution (dark blue) of the network state identified at 0.60 eV. STM parameters: **a** 50 × 50 nm², 100 pA; −100 mV; **b** 6 × 6 nm², 20 pA; −1 V; **e**, **f** Setpoint 100 pA, −1V, $V_{rms}$ = 15.8 mV, $f_{osc}$ = 817 Hz).

respect to the underlying substrate, resulting in unit vectors of 2 nm with first Fe neighbors distanced by 1.15 nm.

The electronic structure of this 2D-MOCN is obtained by means of dI/dV grids from which STS spectra at selected network locations (Fig. 1e) and dI/dV maps (Fig. 1f) are extracted. These STS spectra clearly differ from the uncoordinated DCA (cf. Supplementary Fig. S1). Two broad features call for our attention: The first, in the occupied region close to the Shockley state onset position ($\approx -0.50$ V), and the second at the unoccupied region centered at 0.60 V. In the dI/dV map at $-0.45$ V this first state is rather featureless both on the metal and throughout the network, revealing a substrate origin. This is supported by the fact that no distinct confined state could be detected at the MOCN nanocavity centers, which further supports the weak surface-network interaction. Contrarily, the dI/dV map at 0.60 V shows distinct features throughout the network, primarily at the DCA edges and at the metal centers, as sketched in Fig. 1g. This has been identified as the fingerprint of an extended network multi-band in the related Cu-DCA/Cu(111) system[29].

## Magnetic characterization

Our results evidence perfect crystalline quality and collective electronic states in this 2D-MOCN. Therefore, its magnetic properties can be unveiled using spatially averaging synchrotron-based techniques. A true magnetic signal probed by XAS and XMCD requires that we prevent Fe cluster formation (cf. Supplementary Figs. S2 and S3). Thus, we target untraceable Fe undercoverage samples exclusively leading to

single metal centers surrounded by three N atoms (from different cyano groups) which results in a local three-fold symmetric arrangement ($C_{3v}$). This is a single layer MOCN system, so interlayer coupling is discarded due to the non-magnetic character of the underlying Au substrate. Figure 2a and b show XAS and XMCD spectra respectively, acquired at the $L_{2,3}$ edges of Fe at normal ($\varphi = 0°$) and grazing ($\varphi = 70°$) incidence (see Supplementary Information (S.I.) for experimental details). Both XAS and XMCD show narrow peak contributions, which are reminiscent of systems featuring monodispersed Fe atoms on surfaces[30,31], or embedded in other 2D-MOCNs[17], or forming part of molecules[15]. Indeed, the spectra clearly suggest a Fe(II) oxidation state (see for example ref. 32). To further confirm the absence of Fe cluster formation in the 2D-MOCN, we directly deposit the same amount of Fe on the clean Au(111) substrate (without DCA molecules) and measure Fe $L_{2,3}$ XAS and XMCD under identical experimental conditions. The Fe/Au(111) results are shown in Supplementary Fig. S4. Distinctly different XAS and XMCD spectra are obtained compared to the ones of Fig. 2a which exhibit the typical smoother and broader metallic Fe $L_{2,3}$ lineshapes[31,33]. These spectra evidence that Fe adatoms are assembled into small clusters nucleating at the herringbone elbows (see Supplementary Fig. S2a)[34,35]. A direct inspection of Fig. 2b reveals a strong OOP anisotropy for our 2D-MOCN. Identical results have been obtained in two different XAS/XMCD experimental runs, using different Au(111) crystals as substrate (cf. Supplementary Figs. S5 and S6). Despite a common OOP character, the Fe clusters in the Fe/Au(111) sample show a considerably smaller anisotropy than that found on the

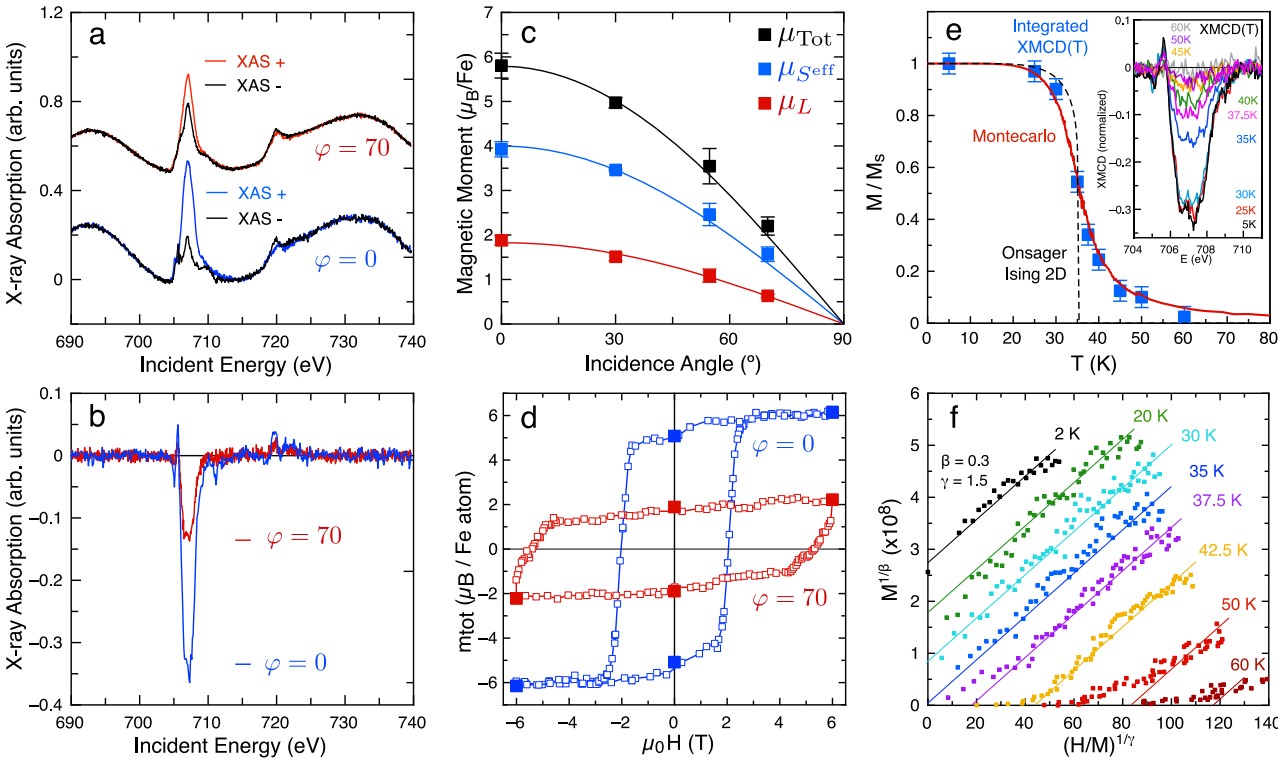

**Fig. 2 | Experimental demonstration of 2D-Ferromagnetism of the Fe-DCA network on Au(111). a** XAS (displaced vertically for clarity) and corresponding (**b**) XMCD spectra acquired with circularly right ($I^+$) and left ($I^-$) polarized X-ray light for normal (0°) and grazing (70°) incidence at the $L_{2,3}$ edges of Fe. The Fe $L_{2,3}$ XAS sits on top of the Au EXAFS background. A strong out-of-plane magnetic anisotropy is evidenced by inspection. **c** Angular dependence of the orbital (red), effective spin (blue) and total (black) magnetic moments obtained from the sum rules. The lines follow a cosine relation: $\mu_{\kappa}^{\varphi=0} \cdot \cos(\varphi)$ with $\kappa = L$, $S_{eff}$, and Total. **d** Hysteresis loops (open symbols) obtained at the $L_3$ edge of Fe at normal ($\varphi = 0°$, blue) and grazing ($\varphi = 70°$, red) incidence. The solid symbols are the result of applying the sum rules to the XAS and XMCD spectra obtained at remanence after conveniently cycling the

field from ±6 T to zero for both incidence angles. **e** Integrated area (normalized to the saturation value) of the $L_3$ XMCD main peak measured in normal incidence under low fields ($H = 0.1$ T) as a function of temperature from $T = 5$ K up to 60 K. The original XMCD data are shown in the inset. The Onsager 2D Ising analytical solution (dashed line) and a Monte Carlo simulation (red line), both for a honeycomb lattice are shown, satisfactorily describing the experimental data for $J \approx 2$ meV. The experiment was performed crossing $T_C$ heating up and then cooling down, exhibiting reversibility. Indeed, such $T_C$ reversibility was reproduced on a second Au(111) substrate under $H = 0.05$ T (cf. Supplementary Fig. S6). **f** Modified Arrott-Noakes plot of isotherms with $\beta = 0.30$ and $\gamma = 1.51$ corresponding to a long-range 2D-Ising model[44].

MOCN, and their $L_3$ vs $L_2$ branching ratio (relative peak intensities) is significantly lower than for the Fe-DCA network, evidencing a much smaller Fe orbital moment. Note that the Fe clusters are not fully comparable to the Fe centers in the 2D-MOCN because their geometry and chemical environments are different. Particularly, the coordination differs not only laterally, but also affects its vertical interaction with the substrate.

We can quantitatively determine the orbital ($\mu_L^z$) and effective spin ($\mu_S^{\text{eff}_z} = \mu_S - 7\mu_T^z$) magnetic moments for the 2D-MOCN and the Fe clusters by using the X-ray magnetic dichroism sum rules[36,37] (described in the Supplementary Section S3) with the XAS and XMCD spectra acquired at 4 different incidence angles ($\varphi = 0°, 30°, 54.7°$, and $70°$). The spectra were obtained in every case with the magnetic field parallel to the beam direction, such that XMCD yields the projection of the Fe magnetic moment along the direction of the applied field, $m(\varphi) = \mathbf{H} \cdot \mathbf{m}/H$. We use the nominal number of holes in the $3d$ band for Fe(II) ($n_h = 4$) in these sum rules. As evidenced in Fig. 2c, $\mu_L^z$ and $\mu_S^{\text{eff}_z}$ are strongly anisotropic, being much larger when the applied field is perpendicular to the 2D-MOCN plane. The obtained values in normal incidence are $\mu_L^z = 1.88 \pm 0.02 \, \mu_B$ and $\mu_S^{\text{eff}_z} = 4.02 \pm 0.04 \, \mu_B$, consistent with a Fe(II) $d^6$ high-spin configuration with L = 2 and S = 2, carrying a large orbital moment $\langle m_z^l \rangle \approx 2\mu_B$, which is responsible for the large magnetic anisotropy. Such high spin configuration agrees with the $2^+$ state of the Fe $2p$ extracted from our XPS data (see Supplementary Section S4 and Supplementary Fig. S7). Similar cases of $d^6$ high-spin strongly uniaxial angular momenta have been observed, such as single Fe(II) ions atop the nitrogen site of the $Cu_2N$ lattice[38] and Co(III) in one-dimensional cobaltate $Ca_3Co_2O_6$[16].

The four incidence angles graphed in Fig. 2c were selected to allow direct separation of the isotropic spin moment, $\mu_S$, from the dipolar term $-7\mu_T^z$. In particular, if the magnetic moments rotate with the field as the angle-dependent experiment is performed, at the so-called magic angle $\varphi = 54.7°$, the intra-atomic dipolar contribution $\mu_T^z$ cancels, allowing a direct measure of $m_S$[15,39] (see Supplementary Section S3). However, this is not the case in Fe-DCA/Au(111) since we find that both $\mu_L^z(\varphi)$ and $\mu_S^{\text{eff}_z}(\varphi)$ vary as $\cos(\varphi)$. Such behavior can only be rationalized if the magnetic moments stay perpendicular to the 2D-MOCN surface even for the extreme case of $\varphi = 70°$ and applied fields of 6T (see Supplementary Fig. S8), thereby evidencing a robust Ising character with OOP quantization axis.

Such marked OOP robustness should stand out when recording hysteresis loops. Therefore, we acquire the magnetization curves at $T = 3$ K by measuring the XMCD intensity at the fixed photon energy of the $L_3$-edge as a function of the applied magnetic field for normal ($\varphi = 0°$) and grazing ($\varphi = 70°$) incidence (see Fig. 2d). Remarkably, the 2D-MOCN presents a square-like open hysteresis loop with a huge coercive field value ($\approx 2.1$ T out-of-plane and $\approx 5.4$ T at $\varphi = 70°$) and a remanence slightly above 80% of the saturation value. Full zero field XMCD spectra at $\varphi = 0°$, and $70°$ were measured directly after saturation with $\mu_0 H = \pm 6$ T, with sum-rules yielding the values shown as solid symbols in Fig. 2d, consistently scaling with the hysteresis loop curve displayed. It is worth noting that the $\varphi = 0°$ hysteresis loop can be calculated from the $\varphi = 70°$ one (and viceversa), by simply scaling the applied field as $H_{0°} = H_{70°} \cos(70°)$ and the magnetization as $M_{0°} = M_{70°}/\cos(70°)$ (see Supplementary Fig. S8). In other words, the OOP component on the hysteresis loops are identical for all measured angles when projecting the applied magnetic field in that direction, evidencing once more the intense uniaxial character of the magnetic moments of Fe(II) in this 2D-MOCN.

To determine whether the open hysteresis loop reflects a slow relaxation single-atom process[40] or a truly 2D ferromagnetic cooperative phase transition, we measure low field ($\mu_0 H = 0.1$ T) XMCD curves as a function of temperature. The spectra at the $L_3$ edge are shown in the inset of Fig. 2e, where we observe a sudden reduction of the XMCD signal occurring around $T_C \approx 35$ K. This critical temperature

is evident in Fig. 2e when plotting the absolute value of the integrated area of the XMCD main peak at the $L_3$ edge (from 705 to 712.5 eV). The 2D Onsager's analytical solution (at zero field)[41] scaled for $T_C = 35$ K is shown for comparison, yielding an exchange interaction of $J = [T_C \cdot k_B \cdot \log(\sqrt{3} + 2)]/2 \approx +1.98$ meV. A Monte Carlo simulation of the spontaneous magnetization under this low field ($\mu_0 H = 0.1$ T) with that particular exchange interaction constant on a simple honeycomb network of Ising spins with nearest neighbor interactions quite satisfactorily fits our experimental data. Note that this exchange constant is considerable when compared with those found in other single-layer 2D-MOCNs ($J \leq 0.27$ meV)[17,22,23,42].

To further demonstrate that a cooperative ferromagnetic phase transition takes place in our 2D-MOCN we used the so-called modified Arrott-Noakes plots[43], which allow to obtain from the isothermal XMCD magnetization curves the magnetization $\beta$ and susceptibility $\gamma$ critical exponents, while determining the critical temperature. The modified Arrot-Noakes plot displayed in Fig. 2f shows the expected temperature-independent slope of the curves at higher fields, neatly showing the cooperative character of the FM below $T_C \approx 35$ K (whose curve satisfactorily crosses the origin) for $\beta = 0.30$ and $\gamma = 1.50$. However, given the XMCD signal to noise ratio, we estimate the uncertainty in the determination of the critical exponents not better than $\pm 15\%$. The obtained exponents are not coincident with the canonical 2D Ising model ($\beta = 1/8$, $\gamma = 7/4$), but are compatible with other well-established cases of strongly uniaxial 2D systems with long-range interactions, such as URhAl ($\beta = 0.287$ and $\gamma = 1.47$)[44]. Considering the low amount of Fe centers present in the system ($\approx 5\%$ of a monolayer), the experimental determination of these critical values, even with this large uncertainty, is a remarkable achievement in itself[45]. In short, all our XMCD datasets present this 2D-MOCN as an archetype example of a two-dimensional ferromagnet at the single-layer limit.

It is worth noting that Fe magnetic moments are strongly dissimilar when comparing Fe-DCA/Au(111) and Fe/Au(111) samples (cf. Fig. 2b and Supplementary Fig. S4a and Supplementary Table S1). Remarkably, the hysteresis loops measured on the Fe clusters (see Supplementary Fig. S4d) are rather similar at $\varphi = 0°$ and $70°$ and do not reach saturation at the highest accessible external field (6 T). Moreover, the butterfly-shape hysteresis found in grazing incidence (closure at 0 T) of Fe/Au(111) is characteristic of paramagnetic (or superparamagnetic) systems with long relaxation times[46,47]. Such differences in the Fe magnetic moments are related to a change of oxidation state of the Fe atoms associated with the $3d$ orbital occupation of the Fe atoms. In the case of the Fe-DCA/Au(111) network, the Fe atom is in a 2+ oxidation state, whereas in the case of Fe/Au(111) it is in a 0 oxidation state (metallic) (see Supplementary Section S4 and Supplementary Fig. S7). The 2+ state is mostly determined by the mechanism of charge transfer from Fe to DCA in Fe-DCA/Au(111), although there is also a sizable charge transfer from the Fe-DCA overlayer to the Au(111) surface (discussed below). Importantly, such a $Fe^{2+}$ state is also supported by semi-empirical multiplet calculations based on the XAS/XMCD spectra (see Supplementary Fig. S9).

## Theory and discussion

To shed light on these results we perform first-principles density functional theory (DFT) calculations (details in Supplementary Section S2). We start by structurally optimizing the Fe-DCA/Au(111) system using its experimentally determined $(4\sqrt{3} \times 4\sqrt{3})R30°$ periodicity and obtain a non-planar MOCN geometry, shown in Fig. 3a, b (see Supplementary Section S2 for the crystal structure details). The vertical distortion affects the cyano groups that bend downwards to the Fe centers. Such buckled geometry has been observed in many other single-layer 2D-MOCN on noble-metal surfaces[13,29,48,49]. Independently of this buckled geometry, the MOCN retains its two-dimensional character with respect to the magnetic properties of interest since the Fe atoms are coplanar, very much like the transition metal atoms in other 2D-systems, e.g., $CrI_3$.

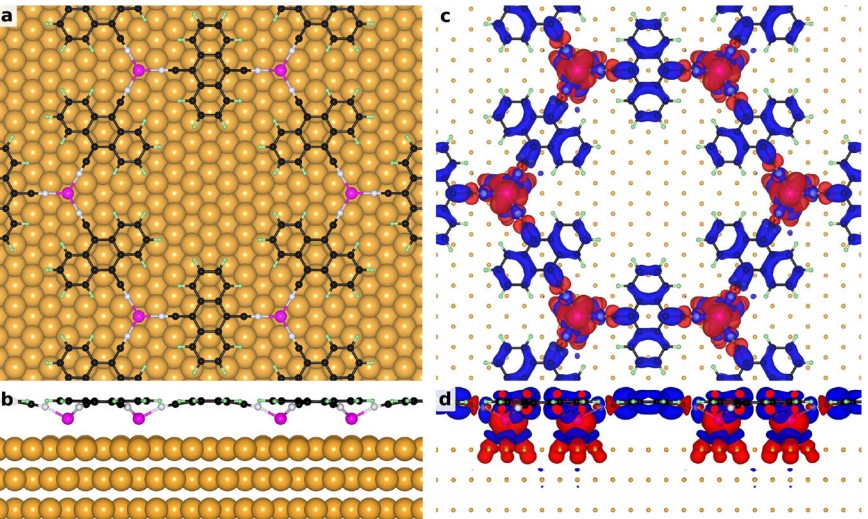

**Fig. 3 | DFT calculations of the Fe-DCA/Au(111) system.** Top (**a**) and side (**b**) views of the 2D-MOCN optimized structure with Au, Fe, C, N and H atoms in gold, purple, black, white, and green colors, respectively. The corresponding spin density iso-contours [red (blue) denotes the majority (minority) spin component] are shown in panels **c** and **d**. This spin density reveals a predominant coupling between Fe magnetic moments through the DCA ligands and a sizable spin polarization of the Au atoms below the Fe atoms. Note that the absence of significant spin density connecting the Au atoms evidences a limited coupling mediated by the metal surface.

Importantly, the shortest Fe-Au distance is about 2.8 Å (see Supplementary Table S5), which is more than 10 % larger compared to the optimized Fe-honeycomb/Au(111) system (i.e. the one relaxed without DCA; see Supplementary Section S5). Thus, the Fe-Au hybridization is quite limited in Fe-DCA/Au(111), as anticipated experimentally. Such hybridization reduction related to the separation of the coordination atom from the substrate is common to other 2D-MOCNs[19,50].

Total-energy calculations of this Fe-DCA/Au(111) structure result in a ferromagnetic isotropic exchange coupling constant of $J \simeq 0.7-1.3$ meV (Hubbard $U_{eff}$ parameter dependent, see Supplementary Table S2). This exchange coupling strength appears to be an order of magnitude weaker when compared to the free-standing planar Fe-DCA system, as discussed in detail in Supplementary Section S5. Noteworthy in this context, is that the buckling of the MOCN induced by the Au(111) substrate makes the Fe-N bond length to be 6–8% longer than in the free-standing Fe-DCA case. This translates into a significantly smaller induced spin polarization in the DCA ligands with respect to that in the free-standing MOCN case, which is consistent with the weaker exchange coupling between the Fe centers. Incidentally, the spin moment of the DCA molecules is opposite in sign to that of Fe, revealing AFM coupling between the Fe centers and DCA ligands, clearly seen in the spin density isosurfaces (Fig. 3c, d). It should be said, however, that the isotropic exchange coupling $J$ between Fe atoms in Fe-DCA/Au(111) is of the same order of magnitude as that between Cr atoms in CrI$_3$[51].

Furthermore, for Fe-DCA/Au(111), we find a positive magnetic anisotropy energy of $E_a \simeq 0.6$ meV per Fe atom (see Supplementary Section S2), indicating an OOP easy magnetization axis, contributions from the single ion anisotropy $D$ and anisotropic exchange interaction $\lambda$ being about $2DS^2 \simeq 1.5$ meV and $3\lambda S^2 \simeq -0.25$ meV (i.e., $E_a \simeq DS^2 + 3\lambda S^2/2$[51]), respectively. Since DFT tends to underestimate the $E_a$ value[52], we have also performed multiplet calculations within a many-body approach based on a point-charge crystal field and find a much larger value of 8.5 meV (Supplementary Section S2 and Supplementary Fig. S10). Despite its quantitative limitations, DFT clearly confirms that the system is a ferromagnet with an OOP orientation of the local magnetic moments of 3.7$\mu_B$ ($S = 2$ state), which nicely matches our experimental results (compare Supplementary Tables S1 and S2 in Supplementary Section S6) and the values obtained from the multiplet calculations. Note that magnetic coupling has been reported in other single-layer 2D-MOCNs directly grown on metals[13,14,17,22–27].

However, in contrast to what is reported here, no open hysteresis loops were detected in any of the previously studied single-layer 2D-MOCNs at the atom thickness limit.

An intriguing question remaining is the definition of the exchange channels that drive the system into such collective magnetic state. From the spin density isosurfaces (Fig. 3c, d) it is seen that there is significant spin density delocalization in the DCA ligands connecting Fe metal centers. Not only the cyano groups, but also the anthracene backbones of DCA appear to be polarized, except for hydrogen atoms, which do not form part of the superexchange coupling path between Fe centers. On the contrary, the spin polarization is only induced on those Au atoms that are the Fe atom's nearest neighbors, which is more clearly seen in the side view (Fig. 3d). Thus, the spin density in Au(111) is quite localized. The fact that no appreciable spin density connects two neighboring Fe centers of Fe-DCA through the Au(111) surface hints to a minor role of the substrate in mediating the exchange coupling between the Fe centers. Therefore, the superexchange via organic ligands dominates in Fe-DCA/Au(111), in agreement with previous works on 2D-MOCN[17,53,54]. However, we do expect some limited coupling mediated by the metal surface, $J_{substr}$, caused by the weak (but still non-negligible) Fe-Au hybridization[55–58]. Regrettably, such coupling mediated by the substrate cannot be separated from that through the ligands in the full Fe-DCA/Au(111) calculation. Therefore, we perform first-principles calculations where we remove the DCA molecules from the optimized system, while keeping the honeycomb Fe array and the underlying Au(111) substrate fixed. Crucially, in this system we fix the Fe-3$d$ manifold occupations to those of Fe in Fe-DCA/Au(111), which is done using the occupation matrix control method. As we discuss in detail in Supplementary Section S5, in spite of being in different chemical environments (Fe-honeycomb-on-Au(111) vs Fe-DCA/Au(111)), the Fe atoms have nearly the same magnetic state. Moreover, the fact that the lateral separation between the Fe atoms as well as their adsorption height above Au(111) turn out to be exactly the same, gives us grounds to use the Fe-honeycomb-on-Au(111) system to estimate the scale of the strength of $J_{substr}$ in Fe-DCA/Au(111). In this way, we find that the substrate contribution to the isotropic exchange is limited to only $J_{substr} \simeq 0.03 - 0.07$ meV (ferromagnetic coupling). Note that this approximation can only provide an order of magnitude estimate because in our calculations the Fe atoms in Fe-honeycomb-on-Au(111) and Fe-DCA/Au(111) are not in an identical state (see Supplementary Section S5).

However, this does not mean that the Au(111) surface plays no role in the observed magnetism of the Fe-DCA MOCN grown on top. Indeed, after formation on the substrate, the network becomes buckled, thereby affecting the Fe-N bond length and hence the induced spin-polarization on the DCA, which in turn influences the main superexchange channel of the Fe-DCA/Au(111). Moreover, there is a charge transfer from the MOCN to the substrate (Supplementary Fig. S11) that affects the Fe-$3d$ occupations (see Supplementary Section S5 and Supplementary Tables S4 and S5), defining the spin magnetic moment.

Summarizing the theoretical part, for this 2D-FM to occur in a MOCN, our first-principles calculations identify two necessary conditions: (i) the existence of hybrid bands (see Supplementary Fig. S11) with magnetic metal centers and organic linker orbital characters responsible for the ferromagnetic coupling between spin magnetic moments in the meV range, and (ii) a large magnetic anisotropy that translates into the opening of a gap in the spin wave excitation spectrum to overcome the Mermin-Wagner theorem[1]. These two key ingredients turn out to be truly remarkable in this Fe-DCA/Au(111) system, causing such finite temperature FM to emerge. Indeed, we find the same order of magnitude in $J$ and roughly equal magnetic anisotropy as in the CrI$_3$ monolayer ($J = 2.2$ meV and $E_a = 0.65$ meV[51]). Thus, it is rather unsurprising that our 2D-MOCN also presents a similar $T_C$ as the one experimentally determined for this 2D-van der Waals ferromagnet ($T_C = 45$ K[9]). At this point, we should mention that a great effort is being done to progress and overcome the challenges involved in tuning the magnetic interactions in metal-organic solids[59,60]. Improving the design of magnetic metal-organic compounds is key to obtain fully functional organic ferromagnets[61,62].

In conclusion, we have studied the structural, electronic and magnetic properties of the Fe-DCA/Au(111) system at its single-layer thickness. This 2D-MOCN exhibits delocalized electronic states with bucked geometry, but limited interaction with the substrate based on the prevalence of the herringbone reconstruction that sets the stage for prevailing electronic overlap between molecules and metal centers and a robust ferromagnetic ground state. Remarkably, we find open hysteresis cycles, an extraordinarily large uniaxial anisotropy, and phase transition behavior which translates into indisputable experimental evidence of a metal-organic 2D ferromagnet with a Curie temperature of $T_C \approx 35$ K. The observed $\mu_L^z/\mu_S^{eff_z} \sim 1/2$ value, consistent with Fe(II) high-spin $d^6$ configuration ($L = S = 2$) is large enough to lead to a high uniaxial magnetic anisotropy (easy OOP magnetization direction). Our first-principles calculations confirm both the order of magnitude of the FM exchange coupling and the sign of the magnetic anisotropy that are necessary to explain the observed magnetic order at a rather high temperature. Importantly, the magnetic exchange constant found for this system is comparable to the highest reported for the ultrathin 2D-van der Waals ferromagnets[10–12] and is certainly much higher than all other previous 2D-MOCNs studied at the single-layer limit[13,14,17,22–27,42].

Our findings settle over two-decades of search for atom-thick 2D-FM in MOCNs, thereby representing a clear advance in the transversal fields of magnetism and surface science. Similarly to the seminal work of isolating a single layer CrI$_3$[9], we expect to boost the community's interest in magnetic 2D-MOCNs and trigger follow-up theoretical and experimental work capable of leading to new ferromagnetic systems that exhibit even higher ordering temperatures with such ultra-low magnetic atom densities.

## Methods

The Methods section, containing experimental and theoretical details, is included in the Supplementary Information file.

## Data availability

The experimental data that support these findings are available from the corresponding authors upon reasonable request.

## Code availability

The theoretical codes that support these findings are available from the corresponding authors upon reasonable request.

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

## Acknowledgements

With this work we pay homage to our college Jorge Cerdà that left us too early, may he rest in peace. We thank the BL-32 staff of ESRF for help in the initial experimental tests and Jesús Bartolomé for valuable help with Monte Carlo simulations. We are also greatful to Guillermo Antorrena, Celia Rogero and Maxim Ilyn for giving us access to their XPS equipments and helping us with the measurements. We further acknowledge the use of Servicio General de Apoyo a la Investigación-SAI and the Laboratorio de Microscopías Avanzadas of the Universidad de Zaragoza. IPZ acknowledges support from Prof. Johannes V. Barth (TUM). We acknowledge financial support from Grant References No. PID2019-107338RB-C64 (J.L.C., L.H.L., A.E.C., D.S.), PID2019-103910GB-I00 (M.M.O., A.A.), PID2020-115159GB-I00 (L.H.L., F.B.), PID2020-116181RB-C32 (P.G., M.V.), PID2022-138750NB-C21 (J.L.C., A.E.C., D.S.), PID2022-138269NB-I00 (F.D.), PTRR-C17.I1 (J.L.C., M.M.O., D.S.), and RED2018-102833-T (J.L.C., D.S.) funded by MCIN/AEI/10.13039/ 501100011033, by "ERDF A way of making Europe" and "European Union NextGenerationEU/ PRTR". Also financing from FlagEra SOgraphMEM PCI2019-111908-2 (P.G., M.V.). (AEI/FEDER) and the European Regional Development Fund (ERDF) under the program Interreg V-A España-Francia-Andorra (Contract No. EFA 194/16 TNSI) (J.L.C., L.H.L., I.P.Z., D.S., F.B.) is acknowledged. We also thank the Basque Goverment Grant No. IT-1527-22 (M.M.O., A.A.) and the Aragonese Projects RASMIA E12_20R (J.L.C., L.H.L., F.B.) and NANOMIDAS E13_20R (D.S.), co-funded by Fondo Social Europeo. The synchrotron

radiation experiments were performed at BOREAS beamline under offical and inhouse proposals ID2020024265 and ID2021095444[45].

## Author contributions

J.L-C., L.H-L. and D.S. conducted the STM experiments and analysis; J.L-C., L.H-L., I.P-Z., A.C., P.G., M.V. and F.B. conducted the XAS / XMCD experiments, and L.H-L. and F.B. analyzed them; J.L.-C. performed the XPS measurements and analysis; F.B. performed the Monte Carlo calculations and the semi-empirical multiplet calculations; M.M.O, J.C. and A.A. performed the DFT calculations; F.D. performed the multiplet calculations; J.L.-C., F.B., M.M.O, A.A., and L.H-L. wrote the manuscript; J.L.-C. and F.B. conceived the project; All authors contributed to the revision and final discussion of the manuscript.

## Competing interests

The authors declare no competing interests.
