## [Peer Review File · Nature Communications]

Reviewers' Comments:

Reviewer #1:

Remarks to the Author:

The manuscript by Lobo-Checa et al. reports the successful synthesis of an interesting Fe-DCA monolayer grown on an Au substrate. The structure and magnetic properties of Fe-DCA monolayer were determined through STM, XMCD, and computational analysis. Notably, Fe-DCA on Au exhibits magnetic ordering temperature of ~35 K with open hysteresis loop due to the strong out-of-plane magnetic anisotropy. Indeed, the isolation of atomically thin layers of magnetic two-dimensional metal–organic frameworks (MOFs) remain highly desirable for many researchers. While in-situ growth and post-synthetic exfoliation techniques had been applied to 2D MOFs, these efforts remain mostly limited to non-magnetic MOFs. This work serves to highlight the active research interest to understand and tune the magnetic properties of MOFs. However, additional experimental evidences are required to justify the conclusion of the manuscript. Upon resolving the issues identified below, this article may be worthy of further consideration for publication in Nature Communications.

1. Typically, in metal–organic coordination compounds, superexchange interaction between spins on metal ions through diamagnetic ligands is weak and observed Curie temperatures (TC) are very low [Chem. Rev. 2020, 120, 8716; Nat. Chem. 2021, 13, 594.]. The use of redox-active ligands and strong antiferromagnetic coupling between spins on metal ions and organic radicals are known to increase the ordering temperature (Neel temperature (TN) for antiferro/ferrimagnets or Tc for critical temperature) [Science 1991, 252, 1415; Science 2020, 370, 587]. In the manuscript, the authors utilized redox-active DCA ligand but lacks a thorough investigation of its redox state. This information is crucial for understanding the magnetism of an Fe-DCA layer. For instance, Fe magnetic spins may have ferromagnetic alignment while the overall Fe-DCA layer exhibits ferrimagnetism due to the antiparallel alignment of DCA radical spins. Additional spectroscopic data comparing DCA in Fe-DCA with non-coordinated DCA in neutral and radical anionic forms are necessary. Some suggestions include XPS measurements [Nat. Chem. 2010, 2, 374; Chem. Mater. 2014, 26, 2883.]

2. Presumably, when the authors deposited Fe on Au substrate, Fe is in 0 oxidation state. In the final Fe-DCA layer, Fe is oxidized and acquires 2+ oxidation state. Please describe the reaction mechanism.

3. The authors claim distortion of Fe ions towards Au surface and hints that this is due to a weak interaction between them (Fe-Au hybridization). However, authors' claim that Au substrate has negligible contribution to the magnetic property of Fe-DCA layer sounds ironic. The authors may have been referring specifically to the exchange interaction mediated by the surface electron on Au substrate. Since the Au layer has obvious impact on determining the coordination environment around Fe ions, it seems erroneous to claim that the observed magnetic property of Fe-DCA/Au arises from Fe-DCA alone. Indeed, it would be ideal to isolate Fe-DCA monolayer or grow Fe-DCA on a non-interacting surface. The authors must, at least mention this limitation in their manuscript. Further, additional computational details must be provided investigating the contribution of Au layer to the overall electronic and magnetic structure of Fe-DCA compound. The authors attempted this through a computational investigation of Fe arrays on Au substrate upon removal of DCA molecules. Since the authors claim that the observed ferromagnetism is intrinsic to a Fe-DCA layer, it seems sensible to provide additional computational results on Fe-DCA compound without the Au substrate. It would be interesting to check if Fe ions acquire three-coordinate trigonal planar (D3h) environment through geometry optimization. If so, it would be extremely valuable to compare results with and without the geometry optimization.

Minor points

1. Strictly speaking, the term "metal–organic framework" is typically for metal–organic coordination

compounds that exhibit permanent porosity. Since Fe-DCA is a monolayer without obvious pore cavity, it is preferable to use other terms, such as metal–organic coordination monolayer. Further, while the authors claim that Fe-DCA exhibits porosity in Results Section A (“evolve into the porous network under...”), the manuscript lacks experimental verification such as adsorption isotherms. The reviewer suggests providing experimental evidences or refrain from using the concept “porosity”.

2. The reviewer suggests removing the last sentence of Results Section A that states, “Such electronic hybridization has also been reported in the Cu+BHT network, revealing superconducting properties”. Benzene hexathiolate is very different from DCA, and Fe-DCA/Au is not superconductor. Please search for other more related transition metal-DCA compounds and use those papers as references.

3. In Results Section C, the authors state “Note that ferromagnetic coupling seems to be rather general in metallic MOFs”. However, there are numerous MOFs with itinerant electrons that have paramagnetic, antiferromagnetic, or ferrimagnetic ground states. Further, the references that the authors included do not adequately support the statement. They are either molecular compounds, lack metallic properties, or magnetic interactions are driven by itinerant electron on the metallic substrate.

4. In Results Section C, the authors claim “No open hysteresis loops were detected in any of the previously studied 2D-MOFs”. While this may be true in metal–organic monolayers, numerous two-dimensional MOFs are known to exhibit open hysteresis loops. This claim may be fine if the authors identify their compound as metal–organic coordination monolayer.

5. In Results Section C, the authors claim “From the spin density isosurfaces it is evident that superexchange via DCAs dominates, in agreement with previous works”, followed by several references. However, all of the references are on very different systems. I suggest looking into coordination solids or molecular compounds of transition metal ions bridged by DCA ligands.

Reviewer #2:

Remarks to the Author:

Lobo-Checa and co-workers reported the ferromagnetic ordering of Fe(II)-spins with Curie temperature of $T_C = 35$ K in a monolayer thick 2D metal-organic framework (2D-MOF) constructed on Au(111) substrate. The manuscript reports the on-surface formation of 2D-MOF through sequential deposition of DCA molecules and Fe-atoms on Au-substrate and subsequently investigated the molecule-substrate interactions using state-of-the-art surface techniques. The magnetic properties were characterized using XAS and XMCD measurements, which reveal ferromagnetic spin-alignments of Fe-3d spin-moments. The experimental observations were supported by the density functional theory calculations (DFT) and Monte-Carlo simulations. The uniaxial magnetic anisotropy, Ising behaviors, and two-dimensional honey-comb spin-textures make the system quite interesting. Despite the ferromagnetic alignment of 2D-MOF, I failed to realize any fundamental breakthrough in this work, thus I have reservations about this work being published in Nature Communication. The ferromagnetic properties reported here for the DCA-based 2D-MOF are only marginally better than the coronene-based 2D-MOF reported by Dong et al. (Nat. Commun. 2018). However, a detailed understanding of the complex nature of the mid/long-range (~ 11.5 Å) on surface ferromagnetic order reported here can make the work quite interesting to a broader audience.

The followings are my concerns about this work.

1. It is not clear whether the out-of-plane movement of Fe-atoms towards the substrate (as depicted in Figure 3b) from the tri-coordinated Fe-DCA plan is because of the Fe-Au chemical bond formation or it's an artifact of 2.55% compressive strain applied into the free-standing lattice parameters to fit

the ($4\sqrt{3} \times 4\sqrt{3}$) Au(111) surface constructions. Thus, validation of the computational model by comparing the experimentally observed structural parameters such as Fe-Fe distances is essential.

2. If the out-of-plane distortion of the Fe-atoms is due to the strong chemical interactions between Fe and Au, which is more likely, then the substrate will contribute to the magnetic properties by mediating the magnetic interactions by conducting electrons of the substrate between the two Fe-atoms via RKKY type of interactions. The strong delocalization of spin-density on the Au-atoms of the adsorbed sites indicates this. The isotropic exchange integrations primarily drive the ferromagnetic order here, my intuition is that the substrate provides an alternative route to the super-exchange mechanism via DCA. The authors seem reluctant to address this issue properly. The way the $J_{\text{substrate}}$ were calculated does not make any sense because the Fe-atoms on Fe/Au(111) and Fe-MOF/Au(111) are not only in the different chemical environments but also in different oxidation states, more rigorous and appropriate method is required to comment on the substrate's contributions.
3. The isotropic magnetic exchange interaction for free-standing MOF has not been reported neither for completely relaxed system nor for the compressed cell with OOP Fe-atoms. A comparison of the isotropic exchange interactions in presence of substrate and without substrate can reveal the crucial role of the substrate in mediating the exchange interactions.
4. The spin-orbit included non-collinear calculations were probably performed for the estimation of the magnetic anisotropy. However, the details of these calculations are missing from the manuscript/SI. Similarly, the detailed information regarding the dipole-dipole interaction parameters as well as the anisotropic exchange interactions are not given. A justification should also be provided if the dispersion interactions were neglected for these calculations.

Reviewer #3:

Remarks to the Author:

The authors present the remarkable finding related to a large magnetic coupling occurring in a Fe-DCA metal organic framework supported on a Au(111) metal surface. By combining scanning tunneling microscopy and X-ray magnetic circular dichroism, they investigated the structural, electronic, and magnetic properties of the network, evincing a ferromagnetic order up to 35 K. Comparison with density functional theory provides additional insight into the coupling mechanism among the metal centers.

In my opinion, the manuscript delivers a very clear message, providing a logical sequence of evidences culminating into the main finding, i.e. the clear signature of magnetic order as inferred from the wide open hysteresis loop. Both STM and X-ray characterizations are complete and thoroughly conducted, which allow a coherent modeling of the observations.

I must say that, although the combination of DCA and metal centers didn't appear particularly new to the research field, the high coercive field and Curie temperature presented for the Fe-DCA in this work are very much unexpected. Such result is a big step forward for the surface science, magnetism, and 2D material fields, and deserves publication in a high impact factor journal.

On a more technical side, I have no comments related to the experimental characterization. The data are solid and of very high quality. On the other hand, I feel that the paper still misses to fully unravel a few aspects related to the magnetic anisotropy of the system. Here below are my comments.

- The authors use density functional theory (DFT) to evaluate the magnetic anisotropy of the framework. This approach is a bit questionable for a system with such a large orbital magnetic moment. It is well known that the DFT approach can largely underestimate the orbital moment in atomic-like systems [see for example Phys. Rev. Lett 115, 237202 (2015)] and consequently it is difficult to believe that the related value obtained for the magnetic anisotropy is quantitatively correct. In addition, the very large anisotropy of the XMCD signal points towards a quite large value of the anisotropy, at least of about one order of magnitude larger than the $E_a \sim 0.6$ meV inferred from DFT. I would rather take the calculated value as a lower bound for the experimental value.

- On the other hand, more reliable information on the magnetic anisotropy could be inferred by comparing the X-ray data with simulated spectra from multiplet calculations, which were shown to provide more precise values for close-to-free-atom systems, as the one shown in this work. The comparison with multiplet calculations would also validate the conclusions of a 3d6 configuration for the Fe atoms and provide more precise values for the spin and orbital moments. The author should consider integrating their results with this theoretical approach.

- The authors find a quite large value of the spin moment by applying the sum rules to the experiment. Considering that, in Fe 3d6, sum rules generally underestimate the experimental value of the spin by up to 20% [see Phys. Rev. B 80, 184410], and that the authors didn't correct for the contribution of the dipolar moment, it seems quite fortuitous that the authors find a value of 4 μ_B , which compares too nicely to a pure $S = 2$ system. I think that, also in this case, the comparison with multiplet calculations could help clarifying the actual value of the spin and the related contribution of these factors to the sum rules value.

- For the values of the beta and gamma exponent obtained from the Arrot-Noakes plot (fig 2f), did the author performed a fit of the data? It would be appropriate to provide these values in the form of best fits and related std. deviation.

- I recommend the authors to provide a reference for the expression of $E_a = DS^2 + 3 \lambda S^2 / 2$

- Finally, I suggest the authors to revise some of their claims of novelty all over the text. For example, on page 3, they state that "2D MOFs have historically failed to explicitly exhibit 2D ferromagnetic remanence". Indeed, the present work is a remarkable step forward for a supported 2D single layer metal-organic framework, however other 2D-MOF have been proved to show ferromagnetic interactions (although not at the single isolated layer level). See for example Nature Commun. 9, 2637 (2018).

REFeree RESPONSE NATURE COMMUNICATIONS

Dear Dr. Bladwell,

We would like to start by thanking you and the three Reviewers for your time and dedication to thoroughly consider our manuscript. We have worked very hard to respond to their criticisms and requests that have contributed to the improvement of its overall quality and clarity. In this major revision we have also complied with the editorial formatting requests. All in all, we are convinced that our manuscript now merits its publication in Nature Communications.

In the following, we do a reasoned point-by-point response to all Reviewers' statements and comments. Note that the text from the Reviewers will be shown in blue, our responses will be in black and any introduced or modified text in the revised version will be indicated in bold red. For reviewing purposes, these changes introduced in the new version of the manuscript and SI will be highlighted in yellow. A list of changes is provided at the end of this response.

=====

Reviewer #1:

The manuscript by Lobo-Checa et al. reports the successful synthesis of an interesting Fe-DCA monolayer grown on an Au substrate. The structure and magnetic properties of Fe-DCA monolayer were determined through STM, XMCD, and computational analysis. Notably, Fe-DCA on Au exhibits magnetic ordering temperature of ~35 K with open hysteresis loop due to the strong out-of-plane magnetic anisotropy. Indeed, the isolation of atomically thin layers of magnetic two-dimensional metal-organic frameworks (MOFs) remain highly desirable for many researchers. While in-situ growth and post-synthetic exfoliation techniques had been applied to 2D MOFs, these efforts remain mostly limited to non-magnetic MOFs. This work serves to highlight the active research interest to understand and tune the magnetic properties of MOFs. However, additional experimental evidences are required to justify the conclusion of the manuscript. Upon resolving the issues identified below, this article may be worthy of further consideration for publication in Nature Communications.

1. Typically, in metal-organic coordination compounds, superexchange interaction between spins on metal ions through diamagnetic ligands is weak and observed Curie temperatures (TC) are very low [Chem. Rev. 2020, 120, 8716; Nat. Chem. 2021, 13, 594.]. The use of redox-active ligands and strong antiferromagnetic coupling between spins on metal ions and organic radicals are known to increase the ordering temperature (Neel temperature (TN) for antiferro/ferrimagnets or Tc for critical temperature) [Science 1991, 252, 1415; Science 2020, 370, 587]. In the manuscript, the authors utilized redox-active DCA ligand but lacks a thorough investigation of its redox state. This information is crucial for understanding the magnetism of an Fe-DCA layer. For instance, Fe magnetic spins may have ferromagnetic alignment while the overall Fe-DCA layer exhibits ferrimagnetism due to the antiparallel alignment of DCA radical spins. Additional spectroscopic data comparing DCA in Fe-DCA with non-coordinated DCA in neutral and radical anionic forms are necessary. Some suggestions include XPS measurements [Nat. Chem. 2010, 2, 374; Chem. Mater. 2014, 26, 2883.]

As the Reviewer points out, the redox state affects the magnetic state of the 2D-framework. However, the question whether the overall Fe-DCA layer exhibits ferrimagnetism due to the antiparallel alignment of DCA radical spins is already answered by the theory. Fig. 3c,d shows in the spin density isocontours that the anthracene backbones of DCA have an opposite spin sign to the Fe metal centers, but the key point is that this is a delocalized state. In other words, after network formation, the metal centers and molecules cannot be considered

independent entities, as we show with the dI/dV maps in Fig. 1f (in agreement with *Nanoscale* **13**, 5216 (2021)). In our Fe-DCA network the total magnetic moment, induced by the Fe atoms in DCA, is equal to about $-0.1 \mu_B$ (the minus sign indicates that the induced moment is antiparallel to that of Fe, see Table R2 in point #3.2).

Nevertheless, we considered the comment from the Reviewer very seriously and attempted XPS measurements two different times. In our first attempt, we grew the samples in our lab chamber at INMA, in Zaragoza (Spain), and transported the samples in a vacuum suitcase to a service XPS dedicated equipment in our institute. Unfortunately, the residual contamination of the XPS chamber (in the $5 \cdot 10^{-8}$ mbar range) due to volatile components from other users strongly affected the corelevel (CL) lineshapes to the point that we had to discard these measurements. In a second attempt, we traveled to the CFM lab, in San Sebastian (Spain) to use a dedicated multichamber ESCA equipped with XPS and LEED under UHV conditions, where samples could be grown *in-situ*. The spectra represented in Fig. R1 correspond to this second attempt, where we used a non-monochromatized Al $K\alpha$ source to acquire the different spectra for the five representative samples. Despite our best efforts, the minute Fe (< 5% of a monolayer (ML)) and DCA amounts are within the detection limit of the XPS equipment used, resulting in very noisy signals. Note that acquisition times were above 1 hour per spectrum (roughly 8 hours per sample). This time had to be restricted since incipient contamination traces in the carbon and oxygen CL lines started to emerge, which critically affected the N1s and Fe2p lineshapes.

From these XPS spectra presented in Fig. R1 we can extract the following findings:

- No oxidation was observed in the datasets based on the absence of an O1s peak in the vicinity of 530 eV.
- For the Fe 2p signal, the largest contribution at the Fe $2p_{3/2}$ position corresponds to the Au(111) substrate that presents a weak Auger signal at that energy (another photon excitation source was experimentally unavailable at the time of the experiment).
- The metallic state (Fe 0) lineshape and energy position was obtained measuring a high Fe coverage spectrum (Metallic Fe) of roughly 2 ML (707.7 eV for the Fe $2p_{3/2}$).
- For the Fe-DCA network sample (Network) the Fe2p signal clearly originates from a different valence state compared to the metallic one previously mentioned. The average energy position of the Fe $2p_{3/2}$ signal in the Network is estimated to be at $E_B \sim 709.3$ eV, which agrees with a Fe 2+ oxidation state, as we claimed based on the XAS spectra represented in Fig. 2a of the main manuscript.
- For the same amount of Fe as in the network (Fe < 5%) – but without molecules – Fe nanodots are formed at the herringbone elbows (see S.I), being the corresponding lineshape similar to that of the metallic state (Fe 0 state).
- The C1s lineshape shows two well defined components when the molecules form the network (at 285.0 eV and 286.2 eV), which we attribute to the difference between the anthracene backbone carbon atoms and the carbon atoms bonded to the N atoms. In the absence of metal coordination, the aggregated islands of DCA feature a dominant peak at 284.7 eV for this C1s component.
- Concerning the N1s signal, we observe an apparent shift of ~ 1.0 eV to higher binding energies upon network formation, as compared to the molecules forming islands (400.1 eV vs 399.1 eV).

Figure R1. XPS of the different core levels acquired with a non-monochromatized Al $K\alpha$ source for five different samples: Fe-DCA network (Network, red spectra), the DCA islands (DCA, blue spectra), Fe as nanodots nucleating at the herringbone elbows (Fe < 5%, black spectra), a high coverage of Fe (Metallic Fe, gray dotted spectra) and the pristine substrate (Au(111), orange spectra). Despite the long acquisition times (above one hour per spectrum) these spectra were at the detection limit of our equipment (see text for details). In the left column the two components of the Fe2p core level are shown at the top, whereas a close up on the Fe2p_{3/2} peak is represented at the bottom for visualization purposes. Due to the noisy condition of the spectra, a smoothing in an energy window of 1.1 eV is performed for the Fe 2p (continuous lines) and the data are offset for clarity. For the C1s, N1s and O1s, the spectra are offset to coincide at the lowest binding energy side. The ticks in the panel mark the estimated peak position energies.

Summarizing these XPS results, we find that the Fe centers in the network are in a non-metallic state and, in agreement with the XAS/XMCD and theory presented in our work, specifically in a 2⁺ condition. Actually, this is not a surprise, as such non-metallic states have been recurrently reported for metal centers incorporated in molecular cores of porphyrins and phthalocyanines directly sitting on metallic surfaces (see for instance T. Shubina et al., JACS (2006) 128, 5644 or F. Buchner ChemPhysChem (2007) 8, 241).

Therefore, we conclude that the N1s state presents a shift to higher binding energies when forming the network, as compared to the case when the same molecules form compact islands without metal coordination. This positive binding energy shift agrees with the work of Zhang et al., Chem. Commun. (2014) 50, 12289 for the

related network of Cu-DCA/Cu(111) (information mentioned in the text, but not shown in the figures), which they interpreted as a negative charge transfer occurring from the molecule to the coordinated adatom. This interpretation contradicts the often observed case of cyano end-groups depleting the electronic charge of the metal centers when forming organometallic bonds with them, as reported for instance by B. Lowe et al., J. Am. Chem. Soc. 2022, 144, 21389. Based on our XPS, we can claim that the Fe centers are in a 2^+ state because they yield their electrons to the cyano groups when forming the organo-metallic bonds. In a simplistic picture disregarding the substrate, as the stoichiometry is 2 Fe atoms to 3 DCA per unit cell, it means that 4 electrons are shared by the three surrounding molecules.

Now, the question is how can the shift of N1s and C1s be positive when the molecules absorb charge from the metal centers after forming the network. The explanation is simple: the network donates charge to the underlying Au(111) substrate. In fact, we confirmed this effect in our DFT calculations, where an upward shift of the DCA LUMO occurs when comparing the free-standing network (see Fig. R2) with the network on the Au(111) surface (new Fig. S11). Moreover, a direct comparison of the bands of these two systems also evidences a significant Fe-DCA hybridization in the spin minority channel close to the Fermi level. In essence the substrate takes charge from the Fe-DCA overlayer after forming the metal-organic network, as compared to the molecular compact islands without metal coordination.

Figure R2. Calculated low-energy band structures of (a,b) the planar free-standing Fe-DCA and (c,d) the full Fe-DCA/Au(111) system. Panels (a,c) and (b,d) correspond to the spin-up and spin-down channels, respectively. Color circles show projections onto the $\text{CN}^- p_z$ (cyan), $\text{Fe } d_{xz}$ (black), d_{yz} (magenta) and d_{z^2} (blue) orbitals. Note the change of the orbital character of the Fe states near the Fermi level upon placing the Fe-DCA network on the substrate. A strong electronic hybridization between the organic ligands and the Fe centers can be seen in the spin-down channel. Gray lines in (c,d) correspond to the slab bands and are essentially the Au states. A clear up-shift of the Fe- and CN^- -derived bands towards the unoccupied states occurs upon presence of the substrate, indicative of a charge transfer from the MOCN to the latter. The results were obtained for the Hubbard parameter $U_{\text{eff}} = 4 \text{ eV}$.

As pointed out by the Referee, this discussion is important, so despite the limited quality of our XPS data, we have included this discussion in the SI in the new section S4 and also added Fig. R1 as the new Fig. S7. Moreover, Fig. R2 is now included as Fig. S11, substituting former Fig. S8.

2. Presumably, when the authors deposited Fe on Au substrate, Fe is in 0 oxidation state. In the final Fe-DCA layer, Fe is oxidized and acquires 2^+ oxidation state. Please describe the reaction mechanism.

The already discussed XPS datasets clearly show these two different oxidation states of the Fe atoms. Importantly, these new XPS datasets objectively agree with the initially presented XAS/XMCD data. In particular, the clustered Fe atoms on the Au surface display a smooth XAS lineshape (Fig. S4a) that corresponds to a metallic non-oxidized state and does not substantially change with incidence angle (Fig. S4c). Contrarily, the Fe-DCA layer exhibits a 2^+ oxidation state based on the agreement between experimental techniques and the new multiplet calculations for a $\text{Fe}(3d^6)$ ion in C_{3v} point group using the Quanty software (M.W. Haverkort et al., Phys. Rev. B 85, 165113 (2012); and Euro. Phys. Lett. 108, 57004 (2014)) with Crispy as interface (M. Retegan, DOI: 10.5281/zenodo.1008184, 2016) (see Fig. R3). Moreover, the large incidence angle differences (Fig. 2a,b) and the cosine behavior of the different magnetic parameters (Fig. 2c) is only expected for a $S=2$ ($4\mu_B$) high-spin configuration, usually assigned to $3d^6$ occupation corresponding to a canonical $2+$ oxidation state.

Figure R3. Comparison between the XAS/XMCD lineshapes corresponding to the experimental (a,b) and multiplet calculated (α,β) Fe-DCA/Au(111) system. The left panels are the ones from Fig. 2(a,b), whereas the right ones have been semi-empirically generated with the Crispy/Quanty software (M.W. Haverkort et al., PRB 85, 165113 (2012); Euro. Phys. Lett. 108, 57004 (2014); M. Retegan, DOI: 10.5281/zenodo.1008184, 2016) assuming a $\text{Fe}2^+(3d^6)$ ion in C_{3v} point group. The good agreement between the two sets of data confidently provides a very strong theoretical support of the 2^+ Fe(II) state in Fe-DCA/Au(111) network. Calculation details: The calculations include spin-orbit coupling, crystal field (CF) effects (described by the empirical parameters $10Dq = 0.75$ eV, $Dt = -0.26$, $Ds = 0.015$), and a slight reduction of the Slater integrals, which are convoluted with a Lorentzian of $\Gamma = 0.46$ eV to account for intrinsic core-hole lifetime broadening and by a Gaussian of $\sigma = 0.3$ eV to account for instrumental broadening.

As already described in the previous comment, the reaction mechanism relates with the significant interactions occurring between the molecular endgroups and the metal centers upon formation of regular metal-organic networks, the most important being charge transfer. In a simple picture the Fe adatoms donate their two s-electrons to the DCA LUMO orbitals due to the strong interaction with the three surrounding cyano groups. Such type of oxidation process of Fe metal atoms directly on top of a noble metal surface has also been recurrently reported for metal centers incorporated in molecular cores of porphyrins and phthalocyanines, where the Fe atoms are coordinated with four N atoms in the macrocycle (see T. Shubina et al., JACS (2006) 128, 5644 or F. Buchner ChemPhysChem (2007) 8, 241).

Notably, this metal center oxidation has also been reported in the case of Cu-DCA (Fuchs et al, J. Phys.: Mater. 3 (2020) 025001 and L. Yan Adv. Funct. Mater. 2021, 2100519), concluding that the Cu atoms lose their $4s^1$ electrons. In the case of Fe-DCA network, the three molecules would share a total of 4 electrons (network stoichiometry: 3 DCA per 2 Fe atoms), but this only holds as long as charge transfer between the network and the substrate is absent. However, in our case, the charge transfer from the Fe-DCA network to the Au(111) substrate introduces an upward shift of the LUMO orbital, according to our DFT calculations and consistent with the XPS data. Therefore, the reaction mechanism is charge transfer, larger from Fe to DCA than from Fe-DCA to Au(111) until an equilibrium is reached. We acknowledge the Reviewer for his/her comments that has allowed us to shed light into these charge transfer processes.

We add this important information in the new version of the manuscript in section S4 of the SI and in the last paragraph of section B of the main manuscript in the following sentences:

Such differences in the Fe magnetic moments are related to a change of oxidation state of the Fe atoms associated with the 3d orbital occupation of the Fe atoms. In the case of the Fe-DCA/Au(111) network, the Fe atom is in a 2+ oxidation state, whereas in the case of Fe/Au(111) it is in a 0 oxidation state (metallic) (see Section S4 and Fig. S7 of the S.I.). The 2+ state is mostly determined by the mechanism of charge transfer from Fe to DCA in Fe-DCA/Au(111), although there is also a sizable charge transfer from the Fe-DCA overlayer to the Au(111) surface (discussed below). Importantly, such a Fe^{2+} state is also supported by semi-empirical multiplet calculations based on the XAS/XMCD spectra (see Fig. S9).

3.1 The authors claim distortion of Fe ions towards Au surface and hints that this is due to a weak interaction between them (Fe-Au hybridization). However, authors' claim that Au substrate has negligible contribution to the magnetic property of Fe-DCA layer sounds ironic. The authors may have been referring specifically to the exchange interaction mediated by the surface electron on Au substrate. Since the Au layer has obvious impact on determining the coordination environment around Fe ions, it seems erroneous to claim that the observed magnetic property of Fe-DCA/Au arises from Fe-DCA alone. Indeed, it would be ideal to isolate Fe-DCA monolayer or grow Fe-DCA on a non-interacting surface. The authors must, at least mention this limitation in their manuscript.

We should stress that we never used the word “negligible” throughout the manuscript, neither we meant that the “Au substrate has negligible contribution to the magnetic property of Fe-DCA”. Instead, we recurrently used the adjective “limited” in reference to the *magnetic coupling* between the Fe centers *mediated by the Au substrate*.

In the following we shall elaborate further on the metal-organic coordination network (MOCN) distortion, related to the strength of the Fe-Au hybridization, and also discuss other presumably less interactive substrates for growing Fe-DCA network, as suggested by the Reviewer.

First, formation of a single layer metal organic network with 3d-atom coordination on a metal surface typically (to our knowledge) results in metallic centers being closer to the surface than the molecular backbones, even when the interaction with the substrate is relatively weak. This general scenario has been recurrently reported in many works, for example, Gambardella et al., Nat. Mater. 8, 189 (2009); Hua et al., J. Phys. Chem. Lett. 12, 3733 (2021); Martín-Fuentes et al., JACS 144, 16034 (2022); Wang et al., PRL 110, 046802 (2013); Matena et al., PRB 90, 125408 (2014); Gao et al., J. Phys. Chem. C 124, 27017 (2020); Pawin et al., Angew. Chem. Int. Ed. 47, 8442 (2008); Schlikum et al., Nano Lett. 7, 3813 (2007); Yang et al., Nanoscale 10, 3769 (2018); Zhao et al., Langmuir 33, 451 (2017), etcetera.

Thus, the Reviewer is correct in that the “Au layer has obvious impact on determining the coordination environment around Fe ions” because it leads to the MOCN distortion, which is expected based on the evidence of the previous paragraph. In our study, this distortion is determined by the DFT structure optimization calculations. As we show in the next comment below (#3.2), the geometric distortion and Fe-Au hybridization affect the Fe-3d states occupation, the Fe local magnetic moment magnitude and the Fe-Fe exchange coupling strength, as compared to those of the free-standing Fe-DCA in its equilibrium structure.

Second, concerning the strength of the interaction (hybridization) between the Fe centers and the Au(111) substrate, we would like to quote here Umbach et al. PRL 109, 267207 (2012): “the presence of the Au(111) herringbone reconstruction underneath the metal-organic layer indicates a weak interaction with the substrate”. In our case, the herringbone reconstruction of the underlying Au(111) is preserved too, suggesting a weak MOCN-substrate interaction, since in the cases of stronger interaction this reconstruction is known to become distorted or even disappear [see for example Bikaljevic et al., ACS Nano 15, 14985 (2021), Cherkez et al., PRB 93, 045432 (2016), Faraggi et al., J. Phys. Chem. C 116, 24558 (2012), Shie et al., JACS 131, 5376 (2009)]. Moreover, in the comment #3.2 below, we show that in the optimized Fe-DCA/Au(111) structure case, the interaction of the Fe atoms with the Au substrate cannot be considered as strong, which eventually translates into a limited (but, of course, non-negligible) Fe-Fe exchange coupling mediated by Au(111).

Following the suggestion by the Reviewer and the Editor, we have attempted to grow this 2D-MOCN on other weakly interacting surfaces. For this, we have chosen topological insulators Bi_2Se_3 and Bi_2Te_3 , which are known to be much less reactive than noble metals. Fig. R4 (left) shows the result of sequentially depositing DCA and Fe on Bi_2Te_3 . The representative images show tiny network patches surrounded by abundant close packed islands of DCA. The brightest features correspond to the clusterized Fe atoms that fail to coordinate with the molecules. More severe things happen when using Bi_2Se_3 as substrate, since no network was observed and only closed packed islands and Fe clusters were detected throughout the surface.

In another attempt, we tried to grow this network on h-BN/Cu(111) as shown in Fig. R4 (right). Unfortunately, we reached a similar result to that of the topological insulator substrates: the DCAs cluster into compact islands that do not coordinate with the Fe metal centers, which aggregate and become embedded within these molecular islands. This happened even when lowering the temperature during metal deposition.

In short, we could only form the Fe-DCA network on Ag(111) and Au(111). Despite our considerable efforts we could not reproduce this MOCN on any other less-interactive non-metal surface. Note that the tiny islands obtained on Bi_2Te_3 are too small to be sensitive when using average techniques, such as XMCD.

Figure R4. STM images of the sequentially deposited DCA molecules followed by Fe deposition on Bi_2Te_3 (left) and on h-BN grown on Cu(111) (right). In both cases the network does not form as the molecules have a tendency to form closed packed islands and the Fe atoms agglomerate into clusters between these molecular patches. Only in the case of Bi_2Te_3 do we detect small patches of the coordination network, which is absent in the case of h-BN/Cu(111). Note that different unsuccessful attempts were made varying the deposition temperature or doing controlled post-annealings.

To comply with the Reviewer's suggestion, we have added the following sentences in the beginning of Section C:

Such buckled geometry has been observed in many other single-layer 2D-MOCN on noble-metal surfaces. [14, 30, 48, 49]. Independently of this buckled geometry, the MOCN retains its two-dimensional character with respect to the magnetic properties of interest since the Fe atoms are coplanar, very much like the transition metal atoms in other 2D-systems, e.g., CrI_3 . Importantly, the shortest Fe-Au distance is **about 2.8 Å** that is **more than 10 %** larger compared to the **optimized Fe-honeycomb/Au(111) system (i.e. the one relaxed without DCA; see S.I. Section S5)**. Thus, the Fe-Au hybridization is quite limited in Fe-DCA/Au(111), as anticipated experimentally. Such hybridization reduction related to the separation of the coordination atom from the substrate is common to other 2D-MOCNs [20, 50].

And also at the end of Section C, before the last paragraph:

However, this does not mean that the Au(111) surface plays no role in the observed magnetism of the Fe-DCA MOCN grown on top. Indeed, after formation on the substrate, the network becomes buckled, thereby affecting the Fe-N bond length and hence the induced spin-polarization on the DCA, which in turn influences the main superexchange channel of the Fe-DCA/Au(111). Moreover, there is a charge transfer from the MOCN to the substrate (Fig. S11) that affects the Fe-3d occupations (see section S5 and Tables S4 and S5 in the S.I.), defining the spin magnetic moment.

Importantly, we have also added the new section S5 to the S.I. describing in detail the role of the Au(111) substrate on the magnetic properties of the 2D-MOCN.

3.2 Further, additional computational details must be provided investigating the contribution of Au layer to the overall electronic and magnetic structure of Fe-DCA compound. The authors attempted this through a computational investigation of Fe arrays on Au substrate upon removal of DCA molecules. Since the authors claim that the observed ferromagnetism is intrinsic to a Fe-DCA layer, it seems sensible to provide additional

computational results on Fe-DCA compound without the Au substrate. It would be interesting to check if Fe ions acquire three-coordinate trigonal planar (D3h) environment through geometry optimization. If so, it would be extremely valuable to compare results with and without the geometry optimization.

We thank the Referee for this comment as he/she is certainly right. In the revised version of the manuscript, we have improved the discussion about the role of the Au(111) substrate in the properties of Fe-DCA/Au(111) using a line of argumentation based on the DFT results, which are now completed with newly performed calculations. Apart from *Fe-DCA/Au(111)* and non-optimized *Fe-honeycomb*/Au(111)* systems discussed in the first submitted version of the manuscript, we now also consider the fully-optimized *free-standing (FS) Fe-DCA* monolayer and the optimized *Fe-honeycomb/Au(111)* arrangement. Note that we use the asterisk (*) to distinguish between the two Fe-honeycomb-on-Au(111) systems. With these four cases, we can now systematically correlate the crystal structures with their magnetism.

Tables R1 and R2 below summarize the magnetic characteristics of the four systems now considered. The *Fe-DCA/Au(111)* is the “full” system extensively described in the originally submitted version of the manuscript. In this case, the Fe centers show local magnetic moments of 3.68 μ_B . The corresponding occupations of the 3d manifold, as well as the magnetic moments induced in the DCA molecules and on the neighboring Au atoms are indicated in the tables.

Table R1. Local magnetic moments on the Fe atoms, m (μ_B/Fe), as well as the occupations of the Fe-3d manifold for the four following systems: full Fe-DCA/Au(111), planar free-standing (FS) Fe-DCA, Fe-honeycomb*/Au(111), and Fe-honeycomb/Au(111). Note that the Fe-3d occupations are very similar in the Fe-DCA/Au(111) and Fe-honeycomb*/Au(111) systems, which is achieved using the occupation matrix control (OMC) method. Given practically equal 3d occupations, the difference in the local magnetic moments between these two cases comes mainly from the Fe-4s states contribution, which is about 0.2 μ_B in Fe-honeycomb*/Au(111), but roughly an order of magnitude smaller in Fe-DCA/Au(111). The results were obtained for the Hubbard parameter $U_{eff} = 4$ eV.

	m (μ_B/Fe)	Spin-up Fe-3d occupations					Spin-down Fe-3d occupations				
		d_{xy}	d_{yz}	d_{z^2}	d_{xz}	$d_{x^2-y^2}$	d_{xy}	d_{yz}	d_{z^2}	d_{xz}	$d_{x^2-y^2}$
Fe-DCA/Au(111)	3.680	0.9594	0.9653	0.9681	0.9927	0.9950	0.0201	0.0288	0.0645	0.0730	0.9449
Planar FS Fe-DCA	3.576	0.9790	0.9545	0.9495	0.9495	0.9790	0.0998	0.0930	0.9170	0.0930	0.0998
Fe-honeycomb*/Au(111)	3.888	0.9554	0.9580	0.9595	0.9596	0.9611	0.0050	0.0236	0.0576	0.0648	0.9501
Fe-honeycomb/Au(111)	3.220	0.9371	0.9381	0.9550	0.9551	0.9555	0.0122	0.0123	0.0180	0.7668	0.7802

Table R2. Isotropic Heisenberg exchange coupling constants J (meV), local magnetic moments on the Fe atoms, m (μ_B/Fe), as well as induced magnetic moments in DCA (where applicable) and on the Au atoms that are Fe’s nearest neighbors (where applicable) for the same systems as in Table R1. Also shown are the d_{Fe-N} , shortest d_{Fe-Au} , and the cyano group d_{C-N} distances. The absolute value of m_{Au} is given for Fe-honeycomb/Au(111) because this system shows antiferromagnetic coupling and, hence, the induced moment sign depends on the Fe local moment direction. Note that when calculating $J = \Delta_{A/F}/6S^2$ for Fe-honeycomb/Au(111) we take $S = 3/2$, since its local magnetic moment is close to 3 μ_B , unlike in other three systems where it is closer to 4 μ_B ($S = 2$). The results were obtained for the Hubbard parameter $U_{eff} = 4$ eV.

	J (meV)	m (μ_B/Fe)	m_{DCA} (μ_B/DCA unit)	m_{Au} (μ_B/Au)	d_{Fe-N} (Å)	d_{Fe-Au} (Å)	d_{C-N} (Å)
Fe-DCA/Au(111)	+1.321 (FM)	3.680	-0.096	+0.01	2.10	2.86	1.18
Planar FS Fe-DCA	+10.663 (FM)	3.576	-0.236	-	1.94	-	1.19
Fe-honeycomb*/Au(111)	+0.038 (FM)	3.888	-	+0.018	-	2.86	-
Fe-honeycomb/Au(111)	-0.340 (AFM)	3.220	-	0.002	-	2.54	-

We now compare *Fe-DCA/Au(111)* to the planar *free-standing Fe-DCA* monolayer. The latter was fully optimized, i.e., the lattice parameter and atomic coordinates correspond to the energy minimum. We found an order of magnitude stronger FM exchange coupling strength compared to *Fe-DCA/Au(111)* (Table R2), as well as different 3d occupations, especially those of the minority $3d_{z^2}$ and $3d_{x^2-y^2}$ orbitals (Table R1). These changes in occupations result in a somewhat smaller spin magnetic moment at the Fe centers ($\sim 3.58 \mu_B$). Nevertheless, a significantly larger spin polarization appears to be induced in the DCA ligands compared to the *Fe-DCA/Au(111)* case (Table R2). The larger induced moment in DCA of the planar *FS Fe-DCA* correlates with the shorter Fe-N bond length as compared to *Fe-DCA/Au(111)*, which is also consistent with the stronger exchange coupling between the Fe centers. Incidentally, the spin moment of the DCA molecules is opposite in sign to that of Fe, revealing AFM coupling between the Fe centers and DCA ligands (i.e., the above mentioned ferrimagnetism, which takes place in both *Fe-DCA/Au(111)* and planar *FS Fe-DCA*). These results already clarify that the origin of the magnetic property of the *Fe-DCA/Au(111)* system lies in the Fe-DCA. Nevertheless, the Au(111) substrate introduces differences between these two systems in the form of network geometry modification, charge transfer and electronic hybridization between the network and the underlying Au(111) surface. The charge transfer can be observed by comparing the electronic band structure near the Fermi level of both systems, i. e., *Fe-DCA/Au(111)* and *FS Fe-DCA* in Fig. R2.

Next, we consider the two auxiliary systems, the non-optimized *Fe-honeycomb*/Au(111)* and the optimized *Fe-honeycomb/Au(111)*, to justify the validity of our estimation of the strength of the exchange coupling J_{substr} mediated by the Au(111) surface, presented in the originally submitted manuscript.

The *Fe-honeycomb*/Au(111)* is an artificial system, derived from the fully-optimized *Fe-DCA/Au(111)* by removing the DCA molecules while leaving the Fe atoms at the same positions. Thus, it forms a 2D Fe honeycomb with an adsorption height of about 2.5 Å above the Au surface. It is important to emphasize that this adsorption height is larger than the equilibrium one, corresponding to the newly calculated *Fe-honeycomb/Au(111)* case. **Crucially, we purposely fix the Fe-3d-manifold occupations** in *Fe-honeycomb*/Au(111)* to those of Fe in the *Fe-DCA/Au(111)* system. This is done using the occupation matrix control (OMC) method (now reflected in the methods section), which is used to pre-converge the charge density using the occupation constraint and then perform a standard DFT calculation without constraints. In this way, we obtain the magnetic configuration of *Fe-honeycomb*/Au(111)* with 3d occupations of the Fe atoms that differ at most by ~ 0.03 (i.e., $\sim 3\%$) from those in the full *Fe-DCA/Au(111)* system (Table R1). As a result, the magnetic moment contributed by the Fe-3d states¹ differs only by $\sim 0.057 \mu_B$ between these two systems. The difference in the Fe *local* magnetic moments m of about $0.2 \mu_B$ between *Fe-honeycomb*/Au(111)* and *Fe-DCA/Au(111)* comes from the magnetization of the Fe-4s states (see caption of Table R1 for more details). The occupation of the Fe-4s states (and hence their magnetization) has not been controlled within OMC. Thus, it is clear that (i) the Fe-3d occupations are practically the same in both *Fe-honeycomb*/Au(111)* and *Fe-DCA/Au(111)*, and (ii) the lateral separation between the Fe atoms as well as their adsorption height above Au(111) turn out to be exactly the same, by construction. Points (i) and (ii) give grounds for the use of this tailored *Fe-honeycomb*/Au(111)* system to estimate the scale of the strength of J_{substr} , which is the exchange coupling mediated by the Au(111) surface in the full *Fe-DCA/Au(111)* system.

As previously described in the originally submitted manuscript, the exchange coupling between Fe spins in this model auxiliary system *Fe-honeycomb*/Au(111)* is more than an order of magnitude weaker than in *Fe-DCA/Au(111)*. This strongly implies that the Fe-Fe superexchange via DCAs dominates over the RKKY-type exchange through Au(111). However, it should be emphasized that the calculation for *Fe-honeycomb*/Au(111)* can only provide an order of magnitude estimate because the Fe atoms in *Fe-honeycomb*/Au(111)* and *Fe-DCA/Au(111)* are not in an identical state.

¹Not to be confused with the local magnetic moment m given in Tables R1 and R2, which contains contributions of *all* Fe valence states

Having compared the strengths of the Fe-Fe exchange via DCAs and via Au(111), one may also compare the estimated J_{substr} to the strongest possible Fe-Fe exchange in this Fe-honeycomb-on-Au(111) for the fixed Fe-Fe lateral separation. To do this, we allow the Fe atoms of *Fe-honeycomb*/Au(111)* to find their equilibrium adsorption height, while keeping their *xy* coordinates fixed. We will refer to the resulting system as *Fe-honeycomb/Au(111)* (without asterisk). As a result of such a structural relaxation, the Fe atoms end up being significantly closer to the substrate (adsorption height of ~ 1.82 Å in comparison to ~ 2.5 Å before relaxation in the *Fe-honeycomb*/Au(111)* system), resulting in a shorter Fe-Au bond length by 12%. A very important conclusion we draw here is that the Fe-Au hybridization in *Fe-honeycomb*/Au(111)* [hence, in Fe-DCA/Au(111)] is rather weak, as we discussed in the initially submitted manuscript.

Furthermore, for the *Fe-honeycomb/Au(111)* we do not fix the occupation of the Fe-3*d* orbitals at all, but instead perform a self-consistent optimization of the electronic degrees of freedom to find the optimal occupancies without constraints. This results in different 3*d* occupation numbers and lower spin moment as compared to *Fe-honeycomb*/Au(111)*. Notably, the J_{substr} absolute value in this case turns out to be an order magnitude larger than in *Fe-honeycomb*/Au(111)* system (see Table R2). However, even under this stronger interaction case, J_{substr} is still several times weaker than in the full *Fe-DCA/Au(111)* system. Note that $J_{substr} < 0$ for *Fe-honeycomb/Au(111)*, i.e., the coupling between the Fe centers is antiferromagnetic. It should be said that in both *Fe-honeycomb/Au(111)* and *Fe-DCA/Au(111)* cases, very small magnetic moments are induced by the Fe atoms at the three Au atoms nearest neighbors (two and three orders of magnitude smaller than the Fe's moment).

Summarizing the above analysis of structure and magnetism of the *Fe-DCA/Au(111)*, planar *FS Fe-DCA Fe-honeycomb*/Au(111)*, and *Fe-honeycomb/Au(111)* systems, we conclude that the role of the Au(111) surface in determining the exchange coupling between Fe spins in *Fe-DCA/Au(111)* via an RKKY-type channel is quite limited. Indeed, the coupling through this channel is more than an order of magnitude weaker compared to the superexchange mediated by the DCA molecules. This result is consistent with the estimated value of the exchange coupling constant $J = 0.1$ meV between Mn and Fe atoms in a long-range magnetic 2D supramolecular Kondo lattice of FePc and MnPc grown on Au(111) [Girovsky et al., Nat. Comm. 8, 15388 (2017)].

However, this does not mean that the Au(111) surface does not play any role in the magnetism of this MOCN. Indeed, after formation on the substrate, the network becomes distorted from its planar geometry inducing changes in the magnetic state of the Fe atoms (3*d* occupation and corresponding spin moment), as well as a weakening of spin polarization of the DCA ligands. In this respect, we must emphasize that the N-Fe bond length increases by about 8% from the free-standing *Fe-DCA* monolayer case to the *Fe-DCA/Au(111)* system. In this way, the role of Au(111) is not negligible in determining the exchange coupling between Fe centers, but it is rather indirect as it introduces shifts in the Fe and DCA levels due to a charge transfer and hybridization, which translates into an "effective" screening of the exchange coupling between Fe centers across the DCA ligands.

We add the following text in the main manuscript in the second paragraph of Section C.

This exchange coupling strength appears to be an order of magnitude weaker when compared to the free-standing planar Fe-DCA system, as discussed in detail in the S.I. Section 5. Noteworthy in this context, is that the buckling of the MOCN induced by the Au(111) substrate makes the Fe-N bond length to be 6-8 % longer than in the free-standing Fe-DCA case. This translates into a significantly smaller induced spin polarization in the DCA ligands with respect to that in the free-standing MOCN case, which is consistent with the weaker exchange coupling between the Fe centers. Incidentally, the spin moment of the DCA molecules is opposite in sign to that of Fe, revealing AFM coupling between the Fe centers and DCA ligands, clearly seen in the spin density isosurfaces (Fig. 3c,d). It should be said, however, that the isotropic exchange coupling J between Fe atoms in Fe-DCA/Au(111) is of the same order of magnitude as that between Cr atoms in CrI3 [51].

Moreover, we add this important discussion as section S5 of the SI.

Minor points

1. Strictly speaking, the term “metal–organic framework” is typically for metal–organic coordination compounds that exhibit permanent porosity. Since Fe-DCA is a monolayer without obvious pore cavity, it is preferable to use other terms, such as metal–organic coordination monolayer. Further, while the authors claim that Fe-DCA exhibits porosity in Results Section A (“evolve into the porous network under...”), the manuscript lacks experimental verification such as adsorption isotherms. The reviewer suggests providing experimental evidences or refrain from using the concept “porosity”.

We believe the Reviewer might be too restrictive on the use of these terms. In the past two decades many authors have employed “metal-organic framework” to refer to single-layer metal-organic coordination networks similar to ours: see for example Kumar et al., *Nano Lett.* 18, 5596 (2018); Kumar et al., *Adv. Funct. Mater.* 2106474 (2021); Gao et al., *J. Phys. Chem. C* 124, 27017 (2020); Liu, Lin *ChemPlusChem* e202200359 (2022); Field et al., *npj Computational Materials* 8:227 (2022), Yan et al., *ACS Nano* 15, 17813 (2021); M. Hua et al., *J. Phys. Chem. Lett.* 12, 3733 (2021); Zhou et al., *Appl. Phys. Lett.* 117, 191601 (2020), etcetera.

Likewise, the term “porous” has been extensively used to describe such single-layer metal-organic coordination networks whenever they were not in a compact configuration: see for instance Pivetta et al, *PRL* 110, 086102 (2013); Piquero-Zulaica et al, *Rev. Mod. Phys.* 94, 045008 (2022); Zhang et al., *ACSnano* 9, 8547 (2015); K. H. Chung et al., *Chem. Commun.* 47, 11492 (2011); Müller et al, *J. Phys.: Condens. Matter* 28, 153003 (2016); Li et al., *ACSnano* 7, 10764 (2013); Écija et al., *ACSnano* 4, 4936 (2010); etcetera.

However, we comply with this request as we understand that this might lead to confusion for part of the broad readership of the journal. **Thus, we swap the term 2D-MOF for 2D-metal-organic coordination network (2D-MOCN) and refrain from the use of porosity throughout the manuscript and SI.**

2. The reviewer suggests removing the last sentence of Results Section A that states, “Such electronic hybridization has also been reported in the Cu+BHT network, revealing superconducting properties”. Benzene hexathiolate is very different from DCA, and Fe-DCA/Au is not superconductor. Please search for other more related transition metal-DCA compounds and use those papers as references.

We agree with the Reviewer and have deleted this sentence.

3. In Results Section C, the authors state “Note that ferromagnetic coupling seems to be rather general in metallic MOFs”. However, there are numerous MOFs with itinerant electrons that have paramagnetic, antiferromagnetic, or ferrimagnetic ground states. Further, the references that the authors included do not adequately support the statement. They are either molecular compounds, lack metallic properties, or magnetic interactions are driven by itinerant electron on the metallic substrate.

We follow the advice of the Reviewer and **change the sentence exchanging “ferromagnetic” by “magnetic” and removing “rather general” so that the statement fits with the results in the quoted references.** Now, it reads:

Note that magnetic coupling has been reported in other single-layer 2D-MOCNs directly grown on metals [14, 15, 18, 23–28]

4. In Results Section C, the authors claim “No open hysteresis loops were detected in any of the previously studied 2D-MOFs”. While this may be true in metal–organic monolayers, numerous two-dimensional MOFs are known to exhibit open hysteresis loops. This claim may be fine if the authors identify their compound as metal–organic coordination monolayer.

In agreement to what was responded previously, **we swap the term 2D-MOF for 2D-MOCN throughout the manuscript**. Thus, as indicated, we can make this claim:

no open hysteresis loops were detected in any of the previously studied single-layer 2D-MOCNs at the atom thickness limit.

5. In Results Section C, the authors claim “From the spin density isosurfaces it is evident that superexchange via DCAs dominates, in agreement with previous works”, followed by several references. However, all of the references are on very different systems. I suggest looking into coordination solids or molecular compounds of transition metal ions bridged by DCA ligands.

We thank the Reviewer for pointing this out. We have decided to **change slightly the sentence by exchanging “DCA” by “organic ligands” to be more general**, but still refer to the 2D-MOCN. In this new version the text now reads:

Therefore, the superexchange via organic ligands dominates in Fe-DCA/Au(111), in agreement with previous works on 2D-MOCN [18, 53, 54].

=====

Reviewer #2

Lobo-Checa and co-workers reported the ferromagnetic ordering of Fe(II)-spins with Curie temperature of $T_C = 35$ K in a monolayer thick 2D metal-organic framework (2D-MOF) constructed on Au(111) substrate. The manuscript reports the on-surface formation of 2D-MOF through sequential deposition of DCA molecules and Fe-atoms on Au-substrate and subsequently investigated the molecule-substrate interactions using state-of-the-art surface techniques. The magnetic properties were characterized using XAS and XMCD measurements, which reveal ferromagnetic spin-alignments of Fe-3d spin-moments. The experimental observations were supported by the density functional theory calculations (DFT) and Monte-Carlo simulations. The uniaxial magnetic anisotropy, Ising behaviors, and two-dimensional honey-comb spin-textures make the system quite interesting. Despite the ferromagnetic alignment of 2D-MOF, I failed to realize any fundamental breakthrough in this work, thus I have reservations about this work being published in Nature Communication. The ferromagnetic properties reported here for the DCA-based 2D-MOF are only marginally better than the coronene-based 2D-MOF reported by Dong et al. (Nat. Commun. 2018). However, a detailed understanding of the complex nature of the mid/long-range (~ 11.5 Å) on surface ferromagnetic order reported here can make the work quite interesting to a broader audience.

We consider the Reviewer's opinion extremely subjective when disregarding the fundamental impact of our work after comparing it to the publication of Dong et al. (Nat. Commun. 2018). Particularly, the system from Dong et al. is a stack of several 2D-layers that collectively add to give a ferromagnetic signal. In other words, the coronene based 2D-MOF is, strictly speaking, not a single layer, but a bulk material consisting of many Fe atomic planes. Thus, our fundamental breakthrough is that the ferromagnetic signal strictly originates from a 2D single Fe plane embedded in an atom-thick 2D-metal-organic coordination network (2D-MOCN). Moreover, the amount of Fe is so

scarce (~5% of a monolayer) that such a tiny signal is technically impossible to detect in even the most sensitive SQUID equipment available.

In addition to this atom-thick single-layer condition, we find remarkable its elevated Curie temperature (T_c) and huge coercive fields (H_c). When compared to the results of Dong et al, the value of H_c is approximately an order of magnitude larger and the value of T_c doubles the other (see Fig. R5). Moreover, the square-shape of the loop cycle of our work is a text-book example of a Ferromagnet, which Dong's work does not exactly reproduce.

Figure R5. Comparison of the H_c (top row) and T_c (bottom row) between our work and Dongs' et al one published previously in Nat. Commun. Note the huge magnetization difference when comparing quantitatively the two systems in the top row (the coronene based 2D-MOF is re-scaled to the one on the left). In the bottom row, one T_c approximately doubles the other one.

Based on these experimental findings we believe our system is exceptional and definitely deserves its publication in a top journal such as Nature Communication.

The followings are my concerns about this work.

1. It is not clear whether the out-of-plane movement of Fe-atoms towards the substrate (as depicted in Figure 3b) from the tri-coordinated Fe-DCA plan is because of the Fe-Au chemical bond formation or it's an artifact of 2.55% compressive strain applied into the free-standing lattice parameters to fit the $(4\sqrt{3} \times 4\sqrt{3})$ Au(111) surface constructions. Thus, validation of the computational model by comparing the experimentally observed structural parameters such as Fe-Fe distances is essential.

The $(4\sqrt{3} \times 4\sqrt{3})R30^\circ$ structure is identified from the LEED patterns and the STM topographic datasets. Therefore, we experimentally find commensuration between substrate and metal organic layer. Based on this experimental input we generate our DFT model using the lattice parameters to define the $(4\sqrt{3} \times 4\sqrt{3})R30^\circ$ supercell in which we optimize the atomic positions of the Fe, DCA and relevant Au atoms, as described in the section S2 of the SI.

Therefore, we conclude that the out-of-plane movement of the Fe atoms towards the surface is not an artifact of the calculations due to the compressive strain in the Fe-DCA overlayer, but it is result of a physical geometrical distortion produced by the observed commensurability with the underlying Au(111) surface.

We try to clarify this point in the text at the beginning of Section C as follows:

We start by structurally optimizing the Fe-DCA/Au(111) system using its **experimentally determined ($4\sqrt{3} \times 4\sqrt{3}$)R30° periodicity** and obtain a non-planar MOCN geometry, **shown in Fig. 3a,b (see S.I. Section S2 for the crystal structure details)**. The vertical distortion affects the cyano groups that bend downwards to the Fe centers. **Such buckled geometry has been observed in many other single-layer 2D-MOCN on noble-metal surfaces [14, 30, 48, 49].**

Besides, we reformulate the corresponding part in the S.I. Section S2 as follows:

A compressive strain of about 2.55 % is needed for its cell to match **the experimentally found ($4\sqrt{3} \times 4\sqrt{3}$)R30° Fe-DCA/Au(111) periodicity**, as the optimized bulk lattice parameter of Au is 2.9153 Å.

2. If the out-of-plane distortion of the Fe-atoms is due to the strong chemical interactions between Fe and Au, which is more likely, then the substrate will contribute to the magnetic properties by mediating the magnetic interactions by conducting electrons of the substrate between the two Fe-atoms via RKKY type of interactions. The strong delocalization of spin-density on the Au-atoms of the adsorbed sites indicates this. The isotropic exchange integrations primarily drive the ferromagnetic order here, my intuition is that the substrate provides an alternative route to the super-exchange mechanism via DCA. The authors seem reluctant to address this issue properly. The way the $J_{\text{substrate}}$ were calculated does not make any sense because the Fe-atoms on Fe/Au(111) and Fe-MOF/Au(111) are not only in the different chemical environments but also in different oxidation states, more rigorous and appropriate method is required to comment on the substrate's contributions.

In this comment, we acknowledge that the Reviewer has raised three critical points: (i) the strength of Fe-Au interaction (hybridization), (ii) the spin density around the Au atoms in the vicinity of the Fe centers, both in the context of the Fe-Fe exchange coupling $J_{\text{substrate}}$ mediated by the Au(111) substrate, and (iii) the validity of the $J_{\text{substrate}}$ estimation reported in the manuscript.

In the following we provide a point-by-point reply to these three important aspects.

(i) Strength of the Fe-Au interaction in Fe-DCA/Au(111)

To gauge the strength of the Fe-Au hybridization in Fe-DCA/Au(111), we have performed additional DFT calculations. Namely, we have considered the non-optimized Fe-honeycomb*/Au(111) system, discussed in the first submitted version of the manuscript, and also the system with optimized adsorption heights of Fe atoms. The resulting system is hereinafter referred to as Fe-honeycomb/Au(111) [i.e., without asterisk (*)].

Let us first recall that Fe-honeycomb*/Au(111) is an artificial system, derived from the fully-optimized Fe-DCA/Au(111) by removing the DCA molecules while leaving the Fe atoms at exactly the same xyz-positions. This corresponds to a honeycomb lattice of Fe atoms with the adsorption height of about 2.5 Å above the Au surface. However, as a result of the structural relaxation, the Fe atoms in the Fe-honeycomb/Au(111) case are located significantly closer to the substrate. Indeed, the adsorption height becomes ~1.82 Å in comparison to ~2.5 Å before relaxation in Fe-honeycomb*/Au(111), resulting in a shorter Fe-Au bond length by 12%. Obviously, the Fe-honeycomb/Au(111) case leads to a stronger Fe-Au interaction regime, while Fe-honeycomb*/Au(111) shows much weaker interaction. Since the Fe atoms of Fe-DCA/Au(111) are located at the same height above the substrate as in Fe-honeycomb*/Au(111), we conclude that the Fe-Au interaction in Fe-DCA/Au(111) must be relatively weak too. For the Fe-Au interaction to be really strong, the Fe-DCA network must be about 0.7 Å closer

to the Au(111) substrate than its optimized separation distance in *Fe-DCA/Au(111)*. Note that this does not imply that the Fe-Au interaction is negligible, as it introduces a distortion in the Fe-DCA network.

This discussion is included in a newly created section S5 of the S.I.

(ii) Spin density around the Au atoms in vicinity of the Fe centers as indicator of the J_{substr} exchange coupling strength

In Fig. 3c,d of the main text it is clear that the spin polarization is **only** induced on those Au atoms that are the Fe atom's nearest neighbors. This is more clearly seen in the side view of Fig. 3d. Thus, the spin density in Au(111) is, in fact, quite localized. Moreover, the fact that no appreciable spin density connects two neighboring Fe centers of *Fe-DCA* through the Au(111) surface hints to a rather limited role of the substrate *in mediating the exchange coupling between the Fe centers of Fe-DCA/Au(111)*.

On the contrary, we find a significant spin density delocalization in the DCA ligands connecting Fe metal centers (Fig. 3c,d). Not only the cyano groups, but also the anthracene backbones of DCA appear to be polarized (except for hydrogen atoms, which do not form part of the superexchange coupling path between Fe centers). This points towards a dominating role of the DCA ligands in mediation of the exchange coupling between Fe atoms.

We would like to stress that with this we do not mean that the Au(111) surface does not play any role in the magnetism of *Fe-DCA/Au(111)*. As we discuss below, the substrate-induced geometric distortion, charge transfer from the MOCN to Au(111), as well as weak Fe-Au hybridization affecting the Fe-3*d* states occupation, the Fe local magnetic moment magnitude and the Fe-Fe exchange coupling strength.

To reflect this discussion, we modify Section C of the main text, page 10, second paragraph as follows:

An intriguing question remaining is the definition of the exchange channels that drive the system into such collective magnetic state. From the spin density isosurfaces (Fig. 3c,d) it is **seen that there is significant spin density delocalization in the DCA ligands connecting Fe metal centers. Not only the cyano groups, but also the anthracene backbones of DCA appear to be polarized, except for hydrogen atoms, which do not form part of the superexchange coupling path between Fe centers. On the contrary, the spin polarization is only induced on those Au atoms that are the Fe atom's nearest neighbors, which is more clearly seen in the side view (Fig. 3d). Thus, the spin density in Au(111) is quite localized. The fact that no appreciable spin density connects two neighboring Fe centers of Fe-DCA through the Au(111) surface hints to a minor role of the substrate in mediating the exchange coupling between the Fe centers. Therefore, the superexchange via organic ligands dominates in Fe-DCA/Au(111), in agreement with previous works on 2D-MOCN [18, 53, 54]. However, we do expect some limited coupling mediated by the metal surface, J_{substr} , caused by the weak (but still non-negligible) Fe-Au hybridization [55–58].**

(iii) Validity of the J_{substr} estimation in the originally submitted manuscript

As the Reviewer points out, a non-negligible Fe-Au interaction enables an RKKY-type exchange coupling channel between Fe centers of *Fe-DCA/Au(111)*. Therefore, a reasonable estimation of J_{substr} , which is the exchange coupling between Fe centers mediated by the substrate, is important. We will now justify why the estimation of J_{substr} reported in the originally submitted manuscript is valid, in spite of being the Fe atoms of *Fe-honeycomb*/Au(111)* [discussed in detail in point (i)] and of *Fe-DCA/Au(111)* “**in the different chemical environments**”, as the Reviewer correctly points out. However, we would like to emphasize that we can only estimate an order of magnitude for the exchange coupling between Fe spins mediated by the Au substrate for these two different systems.

The key reason for the claim of validity of our estimation is that **the Fe-3*d* manifold occupations in Fe-honeycomb*/Au(111) were fixed to those of Fe in the Fe-DCA/Au(111) system**. We admit and regret that we

did not state this explicitly in the originally submitted manuscript. The occupations were fixed using the occupation matrix control (OMC) method (now reflected in the section S2 and S5 of the S.I.), which is used to pre-converge the charge density using the occupation constraint and then perform a standard DFT calculation without constraints. In this way, we obtain the magnetic configuration of *Fe-honeycomb*/Au(111)* with 3d occupations of the Fe atoms that differ at most by ~ 0.03 (i.e., $\sim 3\%$) from those in the full *Fe-DCA/Au(111)* system (see Table R1 earlier in this Reply document). As a result, the magnetic moment contributed by the Fe-3d states (not to be confused with the *local* magnetic moment m given in Tables R1 and R2, which contains contributions of *all* Fe valence states) differs only by $\sim 0.057 \mu_B$ between these two systems. The difference in the Fe local magnetic moments m of about $0.2 \mu_B$ between *Fe-honeycomb*/Au(111)* and *Fe-DCA/Au(111)* comes from the magnetization of the Fe-4s states (see caption of Table R1 for more details). The occupation of the Fe-4s states (and hence their magnetization) cannot be controlled within OMC.

From these calculations, we can safely state that:

- a) the Fe-3d occupations are practically the same in both *Fe-honeycomb*/Au(111)* and *Fe-DCA/Au(111)*. Therefore, in spite of being “in the different chemical environments” in these two systems, the Fe atoms nevertheless do not present “different oxidation states”.
- b) the lateral separation between the Fe atoms as well as their adsorption height above Au(111) are, by construction, exactly the same in these two systems.

The points a) and b) give grounds for the use of this tailored *Fe-honeycomb*/Au(111)* system to estimate the scale of the strength of J_{substr} , which is the exchange coupling mediated by the Au(111) surface in the full *Fe-DCA/Au(111)* system. As previously described in the originally submitted manuscript, the exchange coupling between Fe spins in this model auxiliary system *Fe-honeycomb*/Au(111)* is more than an order of magnitude weaker than in *Fe-DCA/Au(111)*. This implies that the Fe-Fe superexchange via the DCAs dominates over the RKKY-type exchange through Au(111).

It is also instructive to compare the estimated J_{substr} to the strongest possible Fe-Fe exchange in the Fe-honeycomb-on-Au(111) for the fixed Fe-Fe lateral separation. To do this, we allow the Fe atoms of the *Fe-honeycomb*/Au(111)* to find their equilibrium adsorption height, while keeping their xy coordinates fixed. The resulting system has been previously introduced as *Fe-honeycomb/Au(111)* (without asterisk). For this system we do not fix the occupation of the Fe-3d orbitals at all, but instead perform a self-consistent optimization of the electronic degrees of freedom to find the optimal occupancies without constraints. This results in different 3d occupation numbers and lower spin moment as compared to *Fe-honeycomb*/Au(111)* (Table R1). Notably, the J_{substr} absolute value in this case turns out to be an order magnitude larger than in *Fe-honeycomb*/Au(111)* (see Table R2). However, even in this stronger interaction case, J_{substr} is still several times weaker than in the full *Fe-DCA/Au(111)* system. Interestingly, for *Fe-honeycomb/Au(111)* we find that $J_{substr} < 0$, i.e. the coupling between the Fe centers is antiferromagnetic.

Both the *Fe-honeycomb*/Au(111)* and the *Fe-honeycomb/Au(111)* cases exhibit very small magnetic moments at the three Au atoms nearest neighbors below the Fe atom (see Table R2), as compared to the DCA magnetic moments in *Fe-DCA/Au(111)*. Therefore, this observation suggests that the role of the Au(111) surface in determining the exchange coupling between Fe spins via an RKKY-type channel is significantly smaller than the super-exchange mediated by the DCA molecules (see also next point #3 below). This is consistent with the estimated value of the exchange coupling ($J = 0.1$ meV) between Mn and Fe atoms in a long range magnetic 2D supramolecular Kondo lattice of FeFPc and MnPc grown on Au(111) [Girovsky et al., Nat. Comm. 8, 15388 (2017)]. However, we want to insist that it was never our intention to imply that the Au(111) surface does not play any role in this system, as it is relevant in the formation of the 2D metal-organic coordination network, which distorts its planar geometry, thereby inducing changes in the magnetic state of the Fe atoms (3d occupation and corresponding spin moment) and weakening the spin polarization of the DCA ligands. Moreover, we now found a

charge transfer from the MOCN to the substrate (see Fig. R2 earlier in this Reply document) that also affects the Fe-3d occupations, defining the obtained spin magnetic moment.

We add the following sentences in Section C at the end of page 10 in the main text:

Therefore, we perform first-principles calculations where we remove the DCA molecules from the optimized system, while keeping the honeycomb Fe array and the underlying Au(111) substrate fixed. **Crucially, in this system we fix the Fe-3d manifold occupations to those of Fe in Fe-DCA/Au(111), which is done using the occupation matrix control method. As we discuss in detail in section S5 of the S.I., in spite of being in different chemical environments (Fe-honeycomb-on-Au(111) vs Fe-DCA/Au(111)), the Fe atoms have nearly the same magnetic state. Moreover, the fact that the lateral separation between the Fe atoms as well as their adsorption height above Au(111) turn out to be exactly the same, gives us grounds to use the Fe-honeycomb-on-Au(111) system to estimate the scale of the strength of J_{substr} in Fe-DCA/Au(111). In this way, we find that the substrate contribution to the isotropic exchange is limited to only $J_{\text{substr}} \approx 0.03 - 0.07$ meV (ferromagnetic coupling). Note that this approximation can only provide an order of magnitude estimate because in our calculations the Fe atoms in Fe-honeycomb-on-Au(111) and Fe-DCA/Au(111) are not in an identical state (see section S5 of the S.I.).**

In addition, at the end of Section C, before the last paragraph:

However, this does not mean that the Au(111) surface plays no role in the observed magnetism of the Fe-DCA MOCN grown on top. Indeed, after formation on the substrate, the network becomes buckled, thereby affecting the Fe-N bond length and hence the induced spin-polarization on the DCA, which in turn influences the main superexchange channel of the Fe-DCA/Au(111). Moreover, there is a charge transfer from the MOCN to the substrate (Fig. S11) that affects the Fe-3d occupations (see section S5 and Tables S4 and S5 in the S.I.), defining the spin magnetic moment.

Finally, we add this important discussion in section S5 in the SI as well as Fig. R2 as Fig. S11 of S.I.

3. The isotropic magnetic exchange interaction for free-standing MOF has not been reported neither for completely relaxed system nor for the compressed cell with OOP Fe-atoms. A comparison of the isotropic exchange interactions in presence of substrate and without substrate can reveal the crucial role of the substrate in mediating the exchange interactions.

In the revised version, we include new calculated data of the planar *free-standing* (FS) Fe-DCA monolayer, whose properties are now compared to those of Fe-DCA/Au(111). First, the FS Fe-DCA monolayer was fully optimized, i.e., the lattice parameter and atomic coordinates correspond to the energy minimum. For this optimized geometry, we found an order of magnitude stronger FM exchange coupling strength compared to Fe-DCA/Au(111) (Table R2), as well as different 3d occupations, especially those of the minority $3d_{z^2}$ and $3d_{x^2-y^2}$ orbitals (Table R1). These changes in occupations result in a somewhat smaller spin magnetic moments at the Fe centers ($\sim 3.58 \mu_B$). Nevertheless, a significantly larger spin polarization appears to be induced in the DCA ligands compared to the Fe-DCA/Au(111) case (Table R2). The larger induced moment in DCA of the planar FS Fe-DCA correlates with the shorter Fe-N bond length as compared to Fe-DCA/Au(111), which is also consistent with the stronger exchange coupling between the Fe centers. Incidentally, the spin moment of the DCA molecules is opposite in sign to that of Fe, revealing AFM coupling between the Fe centers and DCA ligands in both Fe-DCA/Au(111) and planar FS Fe-DCA. These results indicate that the origin of the magnetic property of the Fe-DCA/Au(111) system lies in the Fe-DCA network. Nevertheless, the Au(111) substrate introduces important differences between these two systems in the form of network geometry modification, charge transfer and electronic hybridization between the network and the underlying Au(111) surface. These latter effects can be observed by comparing the electronic

band structure near the Fermi level of both systems, i. e., *Fe-DCA/Au(111)* and *FS Fe-DCA* in Fig. R2, which is now introduced as Fig. S11.

We have performed new calculations in this new version of the manuscript that we add as tables S4 and S5 in section S6 of the SI. Moreover, we add this discussion in section S5 of the SI to reflect this important point.

In the main text, we add the following sentences in the second paragraph in Section C:

This exchange coupling strength appears to be an order of magnitude weaker when compared to the free-standing planar Fe-DCA system, as discussed in detail in the S.I. Section 5. Noteworthy in this context, is that the buckling of the MOCN induced by the Au(111) substrate makes the Fe-N bond length to be 6-8 % longer than in the free-standing Fe-DCA case. This translates into a significantly smaller induced spin polarization in the DCA ligands with respect to that in the free-standing MOCN case, which is consistent with the weaker exchange coupling between the Fe centers. Incidentally, the spin moment of the DCA molecules is opposite in sign to that of Fe, revealing AFM coupling between the Fe centers and DCA ligands, clearly seen in the spin density isosurfaces (Fig. 3c,d). It should be said, however, that the isotropic exchange coupling J between Fe atoms in *Fe-DCA/Au(111)* is of the same order of magnitude as that between Cr atoms in CrI₃ [51].

4. The spin-orbit included non-colinear calculations were probably performed for the estimation of the magnetic anisotropy. However, the details of these calculations are missing from the manuscript/SI. Similarly, the detailed information regarding the dipole-dipole interaction parameters as well as the anisotropic exchange interactions are not given. A justification should also be provided if the dispersion interactions were neglected for these calculations.

In the originally submitted version there is a paragraph in the S.I. describing the details of the magnetic anisotropy calculations, including those of the dipole-dipole interaction. In particular, the k-mesh and energy tolerance criterion are specified, along with the cutoff for the dipole-dipole interactions. As for the anisotropic exchange interaction λ , it is defined in the Hamiltonian given in the S.I. in Eq. S1. It is also said that “determining λ and D requires a relativistic calculation of the latter configurations (meaning FM and AFM) for both in-plane and out-of-plane moment direction”.

In the revised version, we explicitly include the formulae for λ and D in the S.I. section S2 to make our description even more detailed. Note that although a different system is considered there (namely, CrI₃), the magnetic lattice in our case is exactly the same as that in Ref. [51], i.e., a 2D honeycomb lattice.

Concerning the dispersion interactions, these were indeed neglected in our calculations because the dominating coupling between Fe atoms and Au substrate is not of the van der Waals type. This is supported by the following facts:

(i) The Fe atoms in *Fe-DCA/Au(111)* are relatively close to the Au surface, which translates into a Fe-Au distance of 2.86 Å. Compared to the *Fe-honeycomb/Au(111)* (2.54 Å, see Table R2) the separation is 12% larger, but still close enough to have a weak Fe-Au hybridization (see Fig. R2; added to SI as Fig. S11); (ii) If the coupling was largely of the van der Waals type, neglecting this interaction would prevent the Fe-DCA MOCN from sticking to the substrate in the calculations, which evidently does not occur.

Nevertheless, to demonstrate that the van der Waals interaction is not decisive in *Fe-DCA/Au(111)*, we have performed the geometry optimization of the *Fe-DCA/Au(111)* taking van der Waals corrections into account using the DFT-D3 scheme proposed by Grimme [J. Chem. Phys. 132, 154104 (2010), J. Comput. Chem. 32, 1456 (2011)]. Because of the additional attraction due to the van der Waals forces, the 2D-MOCN gets closer to the surface as

compared to the calculation without van der Waals corrections. As a result, the Fe atom and DCA backbone planes above the Au surface can now be found at ≈ 2.23 Å and ≈ 3.27 Å (compared to 2.5 Å and 3.9 Å without van der Waals corrections), respectively. At that distance, the Fe-N and Fe-Au bond lengths become 1.90% and 2.45% shorter and are equal to 2.06 Å and 2.79 Å, respectively (compared to 2.10 Å and 2.86 Å without van der Waals corrections). Notably, with these geometrical modifications we find that the Fe local magnetic moment undergoes practically no changes upon inclusion of the van der Waals forces during optimization, as this value is found to be $3.65 \mu_B$.

Finally, total-energy calculations have been performed to check the magnetic ordering and anisotropy for this optimized geometry, which included the effect of the van der Waals interactions. We found an isotropic exchange coupling constant of $J = 1.046$ meV and a magnetic anisotropy energy of $E_a = 0.35$ meV ($U_{eff} = 4$ eV). These values are in a good agreement with those presented in the supplementary Table S2, which were previously obtained without taking van der Waals forces into account. Thus, at this level of theoretical treatment (that includes the van der Waals corrections) *Fe-DCA/Au(111)* also shows strong ferromagnetic exchange coupling and robust out-of-plane magnetic anisotropy.

Following this important comment of the Reviewer, in the revised version we take two actions:

1.- We explicitly include the formulae for λ and D in section S2 of the SI to make our description even more detailed.

2.- We include the results from the new calculations with van der Waals interaction for the system *Fe-DCA/Au(111)* in the supplementary Table S2 and briefly discuss its limited effect in the supplementary section S2 of the SI.

=====
Reviewer #3:

The authors present the remarkable finding related to a large magnetic coupling occurring in a Fe-DCA metal organic framework supported on a Au(111) metal surface. By combining scanning tunneling microscopy and X-ray magnetic circular dichroism, they investigated the structural, electronic, and magnetic properties of the network, evincing a ferromagnetic order up to 35 K. Comparison with density functional theory provides additional insight into the coupling mechanism among the metal centers.

In my opinion, the manuscript delivers a very clear message, providing a logical sequence of evidences culminating into the main finding, i.e. the clear signature of magnetic order as inferred from the wide open hysteresis loop. Both STM and X-ray characterizations are complete and thoroughly conducted, which allow a coherent modeling of the observations.

I must say that, although the combination of DCA and metal centers didn't appear particularly new to the research field, the high coercive field and Curie temperature presented for the Fe-DCA in this work are very much unexpected. Such result is a big step forward for the surface science, magnetism, and 2D material fields, and deserves publication in a high impact factor journal.

On a more technical side, I have no comments related to the experimental characterization. The data are solid and of very high quality. On the other hand, I feel that the paper still misses to fully unravel a few aspects related to the magnetic anisotropy of the system. Here below are my comments.

- The authors use density functional theory (DFT) to evaluate the magnetic anisotropy of the framework. This approach is a bit questionable for a system with such a large orbital magnetic moment. It is well known that the DFT approach can largely underestimate the orbital moment in atomic-like systems [see for example Phys. Rev. Lett 115, 237202 (2015)] and consequently it is difficult to believe that the related value obtained for the magnetic anisotropy is quantitatively correct. In addition, the very large anisotropy of the XMCD signal points towards a quite large value of the anisotropy, at least of about one order of magnitude larger than the $E_a \sim 0.6$ meV inferred from DFT. I would rather take the calculated value as a lower bound for the experimental value.

Yes, the Reviewer is right. We agree that the DFT calculated value of the magnetic anisotropy energy could well be one order of magnitude lower than the actual value and, therefore, we now complement our work with multiplet calculations, detailed in the next comment below.

In the revised version of the manuscript we explicitly include the multiplet calculation information in the S.I. We also add the following sentence in the main text right after providing the calculated MAE value:

Since DFT tends to underestimate the E_a value [52], we have also performed multiplet calculations within a many-body approach based on a point-charge crystal field and find a much larger value of 8.5 meV (S.I. Section S2 and Fig. S10). Despite its quantitative limitations, DFT clearly confirms that the system is a ferromagnet with an OOP orientation ...

- On the other hand, more reliable information on the magnetic anisotropy could be inferred by comparing the X-ray data with simulated spectra from multiplet calculations, which were shown to provide more precise values for close-to-free-atom systems, as the one shown in this work. The comparison with multiplet calculations would also validate the conclusions of a $3d^6$ configuration for the Fe atoms and provide more precise values for the spin and orbital moments. The author should consider integrating their results with this theoretical approach.

Following the Reviewer's suggestion, we have performed multiplet calculations for the $Fe(3d^6)$ using a point charge model for the crystal field within a many-body Hamiltonian approach that includes spin-orbit interaction. The details of the calculations are now included in section S2 of the revised S.I. The corresponding low energy excitation spectrum obtained from exact diagonalization of the Hamiltonian is shown in Fig. R6a, which is also included in the S.I. as Fig. S10. The lowest energy $Fe(3d^6)$ multiplet corresponds to $L=2$ and $S=2$ and, as one adiabatically connects spin-orbit coupling, it splits in a lowest energy quintuplet with aligned orbital and spin moments, and another quintuplet with anti-aligned orbital and spin moments. The ground state multiplet corresponds to maximum $S_z=1.97$ and $L_z=1.05$ values of the spin and orbital momentum perpendicular to the surface, the first excited state being around 8.5 meV higher in energy. Therefore, these calculations show that the corresponding magnetic anisotropy energy is an order of magnitude larger than the DFT calculated value, in agreement with the Reviewer's suspicion.

As already indicated, **we add this multiplet calculation to the section S2 of the S.I. and the new supplementary Table S3 and the new Figure S10.**

Figure R6. a) (Left panel) Splitting of the lowest energy multiplet ($S = L = 2$) as one adiabatically connects the spin-orbit coupling up to the free ion value. (Right panel) Splitting of the energy levels with the out-of-plane magnetic field. The values in brackets close to the energy levels indicates the expectation values of S_z and L_z obtained at 6 T. b) Spin and c) orbital magnetic moments in units of the Bohr magneton calculated assuming thermal equilibrium at $T = 4$ K for a magnetic field applied out-of-plane (blue), grazing $\varphi = 70^\circ$ (red) and in-plane (green).

- The authors find a quite large value of the spin moment by applying the sum rules to the experiment. Considering that, in Fe 3d6, sum rules generally underestimate the experimental value of the spin by up to 20% [see Phys. Rev. B 80, 184410], and that the authors didn't correct for the contribution of the dipolar moment, it seems quite fortuitous that the authors find a value of $4 \mu_B$, which compares too nicely to a pure $S = 2$ system. I think that, also in this case, the comparison with multiplet calculations could help clarifying the actual value of the spin and the related contribution of these factors to the sum rules value.

The Reviewer is right again and we admit that this value could have been a little bit overestimated based only on our initial DFT calculations. However, the new multiplet calculations discussed in the previous point confirm that the spin moment value is close to $4 \mu_B$.

Furthermore, we have also performed *semi-empirical multiplet calculations* for the core-level spectra of a Fe($3d^6$) ion in C_{3v} point group using the Quanta software [Haverkort et al., PRB 85, 165113 (2012)] based on our XAS and XMCD dataset with the Crispy interface. The calculated spectra take into consideration the same experimental conditions, i.e., $T=2K$, $H=6T$, two different incidence angles (normal, $\varphi = 0^\circ$, and grazing incidence, $\varphi = 70^\circ$) and the magnetic field always parallel to the incoming X-ray beam. The XAS and XMCD calculations shown on Fig. R7a,b correspond to the case of Fe(II) and include spin-orbit coupling, crystal field (CF) effects, and a slight reduction of the Slater integrals. Simulations were performed with Fe(II) in C_{3v} symmetry, with the strength of the CF described by the empirical parameters $10Dq = 0.75$ eV, $Dt = -0.26$, $Ds = 0.015$ for Fe $^{2+}$. To account for the experimental broadening, these calculations were at the end convoluted by a Lorentzian of width $\Gamma = 0.46$ eV for the $L_{2,3}$ edges that introduces an intrinsic core-hole lifetime broadening and a Gaussian of $\sigma = 0.3$ eV to account for the instrumental broadening.

The agreement between the experimental curves (Fig. R7a,b) and the *semi-empirical multiplet calculations* spectra (Fig. R7a,b) is very satisfactory. Especially when considering that the latter have been obtained by adjustment of

the crystal field parameters with just 3 variables. This means that we did not change any of the charge-transfer parameters introduced by default for the nominal Fe(II) spectrum, other than a slight reduction to 0.7 (from the default 0.8) of the Slater integral F_k . The analysis given by Quanta of the initial Hamiltonian gives $\langle S_z \rangle = -1.98$, and $\langle L_z \rangle = -0.97$, in excellent agreement with the many-body electronic multiplet calculation with point-charges crystal field described in the section S2 of the S.I.

Figure R7. (Identical to Fig. R3) Comparison between the XAS/XMCD lineshapes corresponding to the experimental (a,b) and multiplet calculated (α,β) Fe-DCA/Au(111) system. The left panels (experimental) are the ones from Fig. 2(a,b), whereas the right ones have been semi-empirically generated with the Crispy/Quanta software assuming a $Fe^{2+}(3d^6)$ ion in C_{3v} point group. The good agreement between the two confidently provides a very strong theoretical support of the $2+ Fe(II)$ state in Fe-DCA/Au(111) network. Calculation details: The calculations include spin-orbit coupling, crystal field (CF) effects (described by the empirical parameters $10Dq = 0.75$ eV, $Dt = -0.26$, $Ds = 0.015$), and a slight reduction of the Slater integrals, which are convoluted with a Lorentzian of $\Gamma = 0.46$ eV to account for intrinsic core-hole lifetime broadening and by a Gaussian of $\sigma = 0.3$ eV to account for instrumental broadening.

In essence, we conclude that both multiplet calculations coincide that the Fe ions are in a Fe(II) d^6 high-spin (HS) configuration with $\langle S_z \rangle \sim 2\mu_B$, theoretically confirming what we report in our manuscript. The calculated orbital moment is near $\langle L_z \rangle \sim 1\mu_B$.

We implement these results and discussion in section S2 of the S.I.

Moreover, we add the following sentence in the main text:

DFT clearly confirms that the system is a ferromagnet with an OOP orientation of the local magnetic moments of $3.7\mu_B$ ($S = 2$ state), which nicely matches our experimental results (compare tables S1 and S2 in section S6 of the S.I.) and the values obtained from the multiplet calculations.

- For the values of the beta and gamma exponent obtained from the Arrott-Noakes plot (fig 2f), did the author performed a fit of the data? It would be appropriate to provide these values in the form of best fits and related std. deviation.

We have followed the usual Arrott-Noakes procedure (A. Arrott and J. E. Noakes, Phys. Rev. Lett. 19, 786, 1967), which has already been used with XMCD data as experimental magnetization a number of times (see for example Bedoya-Pinto et al., Science 374 (2021) 616; A. A. Baker et al., Phys.Rev. B 92 (2015) 094420; and A.I. Figueroa et al., J. Magn. Magn. Mater. 422 (2017) 93). However, the small magnitude of the XMCD signal and the time-consuming nature of XMCD measurements prevents us from gathering clean data (noiseless) to determine the critical exponents with higher accuracy. In this way, automatic fitting procedures were not fully successful, and the standard deviation of the exponents had to be provided relying on an *educated estimation* process based on the critical temperature of the system. The Referee is right when pointing out that our estimation of the precision in the determination of the critical exponents was too optimistic. Figure R8 shows that changing critical exponents by $\pm 15\%$ is essentially indistinguishable (center and right figures), with only a small but noticeable deviation in the critical temperature, which was our main control parameter to select the preferred values. However, inspection of the left panel in Fig. R8 (ideal Ising 2D critical exponents) shows that we can rule out $\beta = 1/8$ of the left panel, since the isotherms are no longer parallel and the T_c value is clearly underestimated (below $T=20\text{K}$).

Figure R8. Modified Arrott-Noakes plot of isotherms for different coefficients of β and γ . The center panel corresponds to Fig. 2f, whereas the right graph displays the ideal case for the critical exponents of the Ising 2D model.

Following the Referee's suggestion, we have changed the main text to just claim that our data show the presence of a phase transition whose critical exponents are compatible with those of some previously studied 2D systems, although the quality of the data is not high enough to determine the actual values of the exponents, due to an uncertainty of the order of $\pm 15\%$:

The new text at the end of section B now reads:

To further demonstrate that a cooperative ferromagnetic phase transition takes place in our 2D-MOCN we used the so-called modified Arrott-Noakes plots [43], which allow to obtain from the isothermal XMCD magnetization curves the magnetization β and susceptibility γ critical exponents, while determining the critical temperature. The modified Arrott-Noakes plot displayed in Fig. 2f shows the expected temperature-independent slope of the curves at higher fields, neatly showing the cooperative character of the FM below $T_c \approx 35\text{ K}$ (whose

curve satisfactorily crosses the origin) for $\beta = 0.30$ and $\gamma = 1.50$. However, given the XMCD signal to noise ratio, we estimate the uncertainty in the determination of the critical exponents not better than $\pm 15\%$. The obtained exponents are not coincident with the 'canonical' 2D Ising model ($\beta = 1/8$, $\gamma = 7/4$), but are compatible with other well-established cases of strongly uniaxial 2D systems with long-range interactions, such as URhAl ($\beta = 0.287$ and $\gamma = 1.47$) [44]. Considering the low amount of Fe centers present in the system ($\approx 5\%$ of a monolayer), the experimental determination of these critical values, even with a large uncertainty, is a remarkable achievement in itself [45]. In short, all our XMCD datasets present this 2D-MOCN as an archetype example of a two-dimensional ferromagnet at the single-layer limit.

- I recommend the authors to provide a reference for the expression of $E_a = DS^2 + 3\lambda S^2/2$

This expression is obtained from Eqs.(4) and (6) of Lado & Fernández-Rossier, 2DMater. 4, 035002 (2017), from the total energy differences for in-plane and out-of-plane spin orientations (per unit cell with two Cr atoms) and dividing by two. This reference was already contained in the original manuscript, but as the Reviewer points out was not indicated in this place.

Therefore, **we now give a reference to the expression in Sect. C, which is number [51] of the new revised manuscript.**

- Finally, I suggest the authors to revise some of their claims of novelty all over the text. For example, on page 3, the state that "2D MOFs have historically failed to explicitly exhibit 2D ferromagnetic remanence". Indeed, the present work is a remarkable step forward for a supported 2D single layer metal-organic framework, however other 2D-MOF have been proved to show ferromagnetic interactions (although not at the single isolated layer level). See for example Nature Commun. 9, 2637 (2018).

We agree with the Reviewer and accordingly have revisited the text. We now limit our claims specifically to the case of 2D metal-organic coordination networks at the single-layer limit. For example, **the sentence mentioned by the Reviewer now reads:**

Despite all these, 2D-MOCNs **at the single-layer limit** have historically failed to explicitly exhibit 2D ferromagnetic remanence [14,15,18,23-28].

Changes introduced in this resubmitted version.

- We have introduced new highlighted text to account for the comments of the Reviewers.
- We have added Sections S3 to S5 to discuss the points mentioned by the Reviewers.
- We have introduced new figures S7, S9, S10 and S11 into the S.I. that are now mentioned in the main text of this work.
- We have added three more tables in section S6 of the S.I.
- We have performed new XPS experiments, described in section XPS Results of the S.I.
- We have carried out new calculations (DFT, multiplets, etc.), described at length in S.I.
- We have added a new co-author (Fernando Delgado) as he was responsible for the point-charge electronic multiplet calculations.
- We have changed the acknowledgements.

Reviewer #2:

Remarks to the Author:

The authors have done an excellent job clarifying the doubts and concerns with additional data and modifying the manuscript accordingly. I am convinced of the authors' remarkable achievement in observing a ferromagnetic ordering on the atom-thick metal-organic framework. Indeed, quantitative segregation of contributions of the magnetic exchange interactions between Fe-atoms via Au-substrate and DCA molecule is a complicated task. In this regard, the author's message is quite clear now. I am happy to recommend the article for publication in Nature Communication.

A minor comment:

The appropriate reference should be provided for the added expression for "single-ion anisotropy D" and "anisotropic exchange" in p6 SI.

Reviewer #3:

Remarks to the Author:

The authors addressed my comments in great detail. The inclusion of results from multiplet calculations allowed clarifying my concerns about the values of the magnetic moments and the anisotropy. In addition, these results further validate the assignment of a 3d6 electronic configuration for the Fe atoms, providing an answer to some of the concerns raised by the other Reviewers. I think the manuscript, in its revised form, is now suitable for publication in Nature Communications.

REFeree RESPONSE NATURE COMMUNICATIONS

Dear Dr. Bladwell,

We would like to start by thanking the three Reviewers for their time and dedication and for accepting our manuscript in Nature Communications.

Only two very minor points were suggested that we have included in our manuscript. The text from the Reviewers will be shown in blue and our responses in black.

=====

Reviewer #1:

The authors have adequately addressed previous comments by providing additional experimental and computational data. I believe the authors have also made significant improvement to the manuscript by adding new results and discussions.

Upon the careful review of the manuscript, I have a final suggestion. For readers that are unfamiliar, I strongly suggest adding several sentences that discuss the challenges and progress on tuning magnetic interactions in metal-organic solids or designing magnetic metal-organic compounds (not limited to two-dimensional solids). I also recommend adding more references, including the following four papers (Nature Communications allows up to 70 references): Chem. Rev. 2020, 120, 8716; ACS Cent. Sci. 2023, 9, 777; Nat. Chem. 2021, 13, 594; Science, 2020, 370, 587.

Upon addition of the introductory sentences and references, acceptance is recommended.

We thank the Reviewer for critically reading our work.

Following the advice, we have added these sentences and the new references [59 – 62] in the manuscript.

We are grateful for the acceptance of our manuscript.

=====

Reviewer #2:

The authors have done an excellent job clarifying the doubts and concerns with additional data and modifying the manuscript accordingly. I am convinced of the authors' remarkable achievement in observing a ferromagnetic ordering on the atom-thick metal-organic framework. Indeed, quantitative segregation of contributions of the magnetic exchange interactions between Fe-atoms via Au-substrate and DCA molecule is a complicated task. In this regard, the author's message is quite clear now. I am happy to recommend the article for publication in Nature Communication.

A minor comment:

The appropriate reference should be provided for the added expression for “single-ion anisotropy D” and “anisotropic exchange” in p6 SI.

The missing reference was J. L. Lado and J. Fernández-Rossier, 2D Materials 4, 035002 (2017), which is Ref. 12 in the SI. We add this in the SI as requested.

We sincerely thank the Reviewer for his/her time and acceptance of our manuscript without further changes.

=====

Reviewer #3:

The authors addressed my comments in great detail. The inclusion of results from multiplet calculations allowed clarifying my concerns about the values of the magnetic moments and the anisotropy. In addition, these results further validate the assignment of a 3d6 electronic configuration for the Fe atoms, providing an answer to some of the concerns raised by the other Reviewers. I think the manuscript, in its revised form, is now suitable for publication in Nature Communications.

We sincerely thank the Reviewer for his/her time and accepting our manuscript without further changes.